# Modeling the effects of active wake mixing on wake behavior through large-scale coherent structures

Lawrence Cheung[1], Gopal Yalla[2], Prakash Mohan[3], Alan Hsieh[2], Kenneth Brown[2], Nathaniel deVelder[2], Daniel Houck[2], Marc T. Henry de Frahan[3], Marc Day[3], and Michael Sprague[3]

[1]Sandia National Laboratories, Livermore, CA
[2]Sandia National Laboratories, Albuquerque, NM
[3]National Renewable Energy Laboratory, Golden, CO

**Correspondence:** Lawrence Cheung (lcheung@sandia.gov)

**Abstract.** The use of active wake mixing (AWM) to mitigate downstream turbine wakes has created new opportunities for reducing power losses in wind farms. However, many current analytical or semi-empirical wake models do not capture the flow instabilities that are excited through the blade pitch actuation. In this work, we develop a framework, which accounts for the impacts of the large-scale coherent structures and turbulence on the mean flow, for modeling AWM . The framework uses a triple-decomposition approach for the unsteady flow field and models the mean flow and fine-scale turbulence with a parabolized Reynolds Averaged Navier-Stokes (RANS) system. The wave components are modeled using a simplified spatial linear stability formulation that captures the growth and evolution of the coherent structures. Comparisons with high fidelity large eddy simulations (LES) of the turbine wakes showed that this framework was able to capture the additional wake mixing and faster wake recovery in the far wake regions for both the pulse and helix AWM strategies with minimal computational expense. In the near wake region, some differences are observed in both the RANS velocity profiles and initial growth of the large-scale structures, which may be due to some simplifying assumptions used in the model.

## 1 Introduction

Wake propagation downstream of turbines in a wind farm is often dominated by the evolution of large-scale coherent structures. These features may arise from unsteady atmospheric conditions, such as the naturally occurring streaks in an atmospheric boundary layer (Zhang and Stevens, 2020) or result from various control strategies intentionally applied to the turbines upwind. Given the wide range of spatial and temporal scales involved, reduced order models are required to efficiently capture the evolution of turbine wakes and enable evaluation of the impact and performance of various wake control strategies at the farm scale. Early models for this purpose were based on steady-state flow assumptions and are thus unable capture critical dynamic aspects of wake evolution. In this paper, we develop an improved reduced order wake model that incorporates time dependent propagation physics. We show that this new model can be used to more effectively capture wake perturbation and recovery dynamics. Although we focus the present work on capturing coherent structures generated by the turbine control strategies, our methods are likely to be relevant to other time-dependent sources as well.

Wind farm flow control methods are designed primarily to reduce power losses in wind farms due to the effects of wakes on downstream turbines. Common approaches include static or dynamic adjustments to the induction factor (turbine derating), yaw angle (wake steering), or blade pitch (wake mixing) (Meyers et al., 2022) of upstream turbines. The present work focuses specifically on active wake mixing (AWM), which aims to excite flow instabilities in the wake that enhance the entrainment of mean velocity, thereby accelerating wake recovery.

Following Cheung et al. (2024a), AWM can be implemented by specifying a dynamic blade pitch, $\theta(t)$, on top of the baseline pitch set point, $\theta_0(t)$, as,

$$\theta(t) = \theta_0(t) + A\cos(\omega_e t - n\psi(t) + \psi_{\text{clock}}), \tag{1}$$

where $A$ is the pitching amplitude, $\omega_e$ is the excitation frequency, $\psi$ is the azimuth position of the blade, $\psi_{\text{clock}}$ is the clocking angle, and $n$ is an azimuthal wavenumber. The parameter $n$ controls the structure of the flow instabilities imparted on the wake and is often used to distinguish between different AWM strategies. Examples include the pulse method ($n = 0$), which generates an axisymmetric instability in the flow through collective blade pitching (Goit and Meyers, 2015; Munters and Meyers, 2018), and the helix method ($n = -1$), which uses individual pitch control to impart a helical structure on the wake that rotates in the direction opposite the turbine rotor (Frederik et al., 2020a). The instabilities are actuated in the wake according to an excitation frequency, which can be specified as a function of the Strouhal number, $St$, the inflow velocity, $U_{inf}$, and turbine diameter, $D$, as $\omega_e = 2\pi St U_{inf}/D$. Strouhal numbers based on the natural unsteady properties of the wake are typically sought ($St \approx 0.3$), leading to flow structures that are generated over much longer periods than a rotor period (Frederik et al., 2020b).

One type of existing reduced-order wake model is the steady-state, analytical one. This type of model finds its roots in the Jensen (Jensen, 1983) and Ainslie (Ainslie, 1988) models, for instance, and also includes more recent, sophisticated versions such as the cumulative curl model (Bastankhah and Porté-Agel, 2014). These models are available for optimizing wind farm performance through the FLORIS code (Sinner and Fleming, 2024). However, the inherently steady-state nature of their implementations in FLORIS, along with their reliance on empirical tuning limits their applicability in scenarios where unsteady flow features are critical. Recent work has shown the importance of unsteady flow features for AWM by connecting the performance of different AWM strategies to the underlying fluid mechanics associated with the induced flow instabilities, particularly the interactions between unsteady coherent flow structures and wake recovery dynamics (Korb et al., 2023; Cheung et al., 2024a). Notably, Cheung et al. (2024a) introduced a spatial linear stability analysis to quantify the growth characteristics of initial flow disturbances based on the temporal forcing frequency and forced azimuthal wavenumber, and showed a correlation between turbulent entrainment statistics in the wake and modal energy gain. These findings suggest that an accurate model for AWM should be capable of representing the unsteady effects of coherent structures on the flow.

Although wind farm optimization typically relies on steady-state models, several approaches for dynamic wake modeling have also been developed. These methods often rely on data-driven representation of the coherent flow structures. Proper Orthogonal Decomposition (POD), for instance, is a commonly used data analysis technique for identifying the energetic structures in a flow (Lumley, 1967), and it has been applied to a wide range of applications including characterizing the coherent structures in a wind farm (Bastine et al., 2015; Ali et al., 2017; Hamilton et al., 2018; Zhang and Stevens, 2020).

In the context of AWM, Yalla et al. (2025) demonstrated that Spectral POD provides a useful representation of the coherent structures induced by dynamic blade pitch actuation, connecting the frequency and wavenumber inputs used by the turbine controller to structures in the wake. Spectral POD is closely related to other data-driven reduced order modeling techniques such as resolvent analysis and Dynamic Mode Decomposition (DMD) (Towne et al., 2018). Li and Yang (2024) developed a resolvent-based model to represent the wake of floating offshore wind turbines subjected to dynamic platform motions, which produce similar wake responses as the pulse and helix forcing strategies. Gutknecht et al. (2023) developed an AWM DMD model for the wake of a single actuated turbine, which easily scaled to different wind speeds and forcing Strouhal numbers. While these data-driven methods are valuable, they can require a substantial amount of training data to provide accurate representations of coherent flow structures. Generating this data can be prohibitively expensive, and avoiding this expense is a primary motivation for reduced order modeling. Moreover, this reliance on training data can limit the applicability of these models outside the specific conditions for which they were developed, and can make adopting these models challenging. Although other dynamic modeling approaches, such as the Dynamic Wake Meandering model (Larsen et al., 2007; Madsen et al., 2010) exist, they too often rely on external turbulence simulations to provide the dynamic components of the flow. In this work, we propose an analytical representation of the coherent flow structures in the wake and limit the training to model constants, which should enhance the robustness and adaptability of the model.

The behavior of large-scale coherent structures in various canonical shear flows is a well-studied problem with a vast body of existing literature. A number of previous works describe the formation and behavior of these structures in turbulent boundary layers (Hussain, 1986; Robinson et al., 1991), free shear layers (Ho and Huerre, 1984), jets (Crow and Champagne, 1971), and wakes (Fuchs et al., 1979). Of particular relevance to the current work are the theoretical and modeling approaches used to analyze such flows. Hussain and Reynolds (1970) introduced the concept of a triple-decomposition analysis to separate the mean flow, fine-scale turbulent components, and wave components of flow, which was widely used in modeling jet (Iqbal and Thomas, 2007) and boundary layer (Kwon et al., 2016) flows. A number of previous studies have shown that the growth of the coherent structures in shear flows can be modeled by spatial stability theory (Cheung and Lele, 2009; Cheung and Zaki, 2011), and noted that the coupling of the growth of the large-scale structures to the mean flow evolution was critical to capturing the behavior of the flow. However, these modeling approaches have yet to be applied to the problem of turbine wakes, leading to a large gap between the currently available steady-state wake models and computationally expensive, high fidelity simulations.

The objective of the current work is to develop a physics-based, computationally efficient model that can capture the effects of active wake mixing on turbine wakes. While turbine wakes contain significant differences from the canonical jet flows discussed above, we show that, by using a triple-decomposition approach, we can still capture the mean flow using a parabolized RANS model, and the large-scale structures can be modeled with a spatial linear stability formulation. In the following sections, we describe the mathematical formulation used in this study and the high fidelity numerical simulations of the turbine wakes used to calibrate and evaluate the reduced order model. We then show comparisons between LES calculations and the RANS with linear stability model for different AWM strategies, and conclude with a summary of the work and a discussion of future work in this area.

**Table 1.** Hub-height wind speed conditions used in the turbine wake study. All values are taken from the simulated atmospheric boundary layer.

| Name | Wind-Speed (WS) | Turb. intensity (TI) | Shear Exponent | Rotor disk veer |
|------|-----------------|----------------------|----------------|-----------------|
| Low WS | 6.52 m/s | 0.036 | 0.142 | 7.9° |
| Med WS | 9.05 m/s | 0.031 | 0.160 | 8.9° |
| High WS | 11.58 m/s | 0.035 | 0.156 | 5.6° |

## 2 Methodology

### 2.1 Atmospheric and turbine conditions of interest

Though the current model is meant to be broadly applicable to all turbine wake flows from both onshore and offshore horizontal axis wind turbines, this work focuses on modeling AWM as applied to larger offshore wind turbines under low turbulence, relatively steady atmospheric conditions. In these situations, the application of AWM can potentially lead to substantial wake benefits and noticeable AEP gains. For offshore locations, the prevalence of these conditions can also lead to many situations where the turbine wakes are especially long and provides an opportunity to improve wind farm power performance.

Representative offshore conditions were selected for this study based on measured data from a floating lidar measurement campaign conducted off the NY bight (Mason, 2022; DNV, 2023). The floating lidar data, collected in 10-minute intervals over a period of 1.6 years, provided velocity and turbulence intensity (TI) profile information for heights between 20 m and 200 m. From this data, a selection process was undertaken to generate three representative wind speed profiles with relatively low TI (see table 1). To generate the precursor simulations, small velocity and temperature perturbations were introduced near the surface to accelerate turbulence development. The low TI conditions were produced by imposing negative ground surface temperature rates and adjusting the surface roughnesses, followed by 10,000 s of flow time. As such, the generated conditions were stable atmospheric boundary layers. More details of the precursor generation process and comparison versus the measured atmospheric data are described in Brown et al. (2025).

These conditions corresponded to the likely operating range of the IEA 15 MW reference turbine where AWM strategies might be deployed. The IEA 15 MW reference turbine was used in this study due to its similarity to current offshore wind turbines being developed by major turbine OEM's. The details of this turbine are summarized in Table 2, with additional information available from Gaertner et al. (2020).

As discussed in section 2.5.1, the selected wind conditions and turbine model were used to set up high fidelity LES calculations for turbines with and without AWM activated. The LES data were then used to calibrate RANS closure model coefficients in the wake model and to compare the accuracy of the final outputs from the model.

**Table 2.** Details of the IEA 15 MW reference turbine

| Turbine Parameter | Value |
|---|---|
| Hub height | 150 m |
| Rotor diameter | 240 m |
| Rated wind speed | 10.59 m/s |
| Design $C_t$ | 0.804 |
| Design TSR | 9.0 |

## 2.2 Mathematical formulation

To model both the steady-state wake profiles and the unsteady dynamics of coherent structures that may be excited through AWM, we use the triple-decomposition approach pioneered by Hussain and Reynolds (1970) in their studies of boundary layer flows. The triple-decomposition formulation separates the flow velocity $\mathbf{u}(\mathbf{x},t)$ into the three components,

$$\mathbf{u}(\mathbf{x},t) = \overline{\mathbf{U}}(\mathbf{x}) + \tilde{\mathbf{u}}(\mathbf{x},t) + \mathbf{u}'(\mathbf{x},t), \tag{2}$$

where $\overline{\mathbf{U}}(\mathbf{x})$ is the time-averaged mean flow, $\tilde{\mathbf{u}}(\mathbf{x},t)$ is the wave component of the velocity, and $\mathbf{u}'(\mathbf{x},t)$ are the fine scale
turbulent fluctuations. The mathematical operations required to compute the mean flow over an averaging time, $T$, are given by

$$\overline{f(\mathbf{x})} = \frac{1}{T} \int_0^T f(\mathbf{x},t) \, \mathrm{d}t, \tag{3}$$

and the phase average is defined by

$$\langle f \rangle = \frac{1}{N} \sum_{m=0}^N f(\mathbf{x},t+m\tau), \tag{4}$$

for a given time period, $\tau$, of the coherent structure and for a specified number of periods, $N$. Once the mean and the phase averaged velocities are known, the wave component, $\tilde{\mathbf{u}}(x,t)$, of the flow field is defined as

$$\tilde{\mathbf{u}}(x,t) = \langle \mathbf{u}(x,t) \rangle - \overline{\mathbf{U}}(\mathbf{x}), \tag{5}$$

and the fine scale fluctuating components can be calculated as

$$\mathbf{u}'(\mathbf{x},t) = \mathbf{u}(\mathbf{x},t) - \overline{\mathbf{U}}(\mathbf{x}) - \tilde{\mathbf{u}}(\mathbf{x},t). \tag{6}$$

An example of a turbine wake that has been triply-decomposed is shown in Figure 1. In this case, the turbine flow field was calculated using LES and averaged according to definitions (3) and (4), which leads to clear depictions of the mean flow field features as well as the large-scale coherent structures that develop within the wake. In the current study, an averaging time of $T$=600s is typically applied to the unsteady LES data.

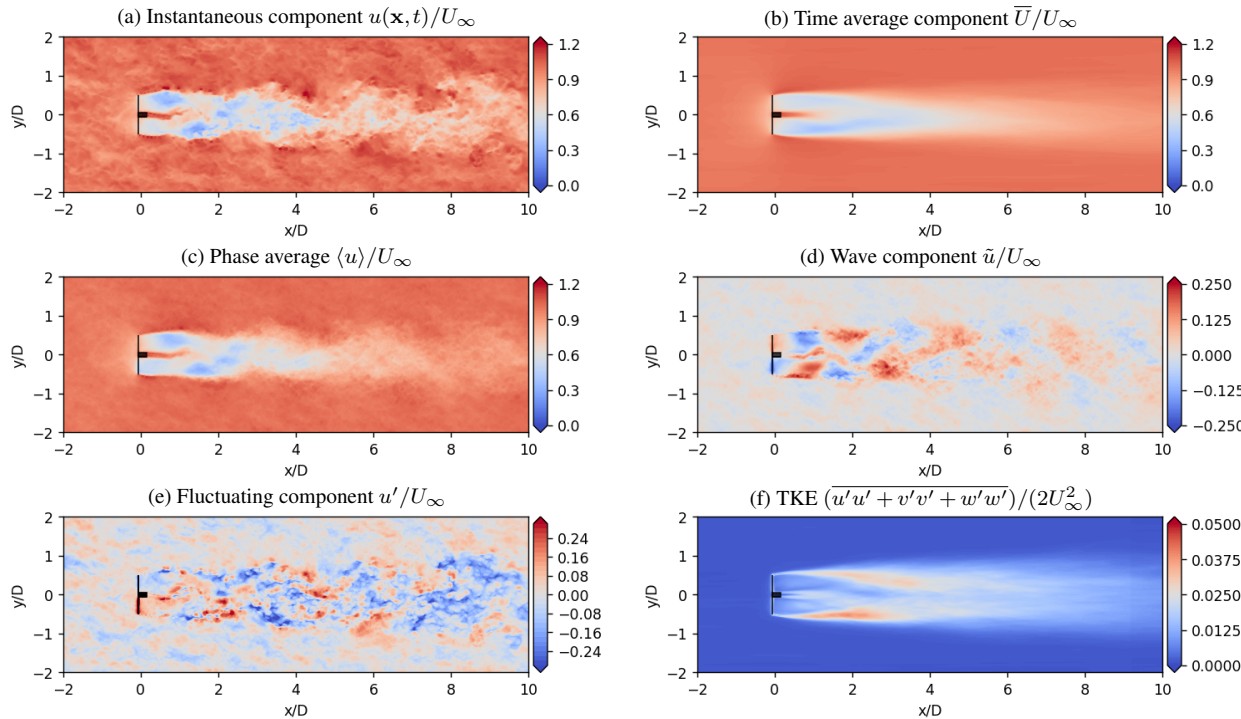

**Figure 1.** An example of a triply decomposed flow field for a wind turbine wake. This case is from the HelixA4 case in the Low WS ABL conditions with $4°$ amplitude forcing. In each contour, the normalized streamwise velocity, $U/U_\infty$, is plotted.

One advantage of using the triple-decomposition approach is that it allows for computationally efficient models to be developed that can solve for each of the three components. Interactions among the different flow components can also be included, which shows how the large-scale coherent structures can impact the mean flow and vice-versa. In the following sections, we describe how a parabolic RANS model can be used to efficiently capture the mean flow and fine-scale turbulent flow components. This is coupled to a linear stability model for the wave components of the flow, and we show that, as the large-scale coherent structures develop within the wake, the mean velocity profiles are impacted as well, leading to the desired wake mixing behavior in this application.

The current model described in the work applies to the wake, immediately downstream of the rotor, of a single turbine. The unsteady inflow effects and the rotor loading dynamics are not explicitly in this formulation, and the behavior of more complicated phenomena, such as the merging of multiple wakes, is not considered here. With additional development, we intend to extend the current model to wind farm configurations with multiple turbines, but we focus on the single turbine wake dynamics initially.

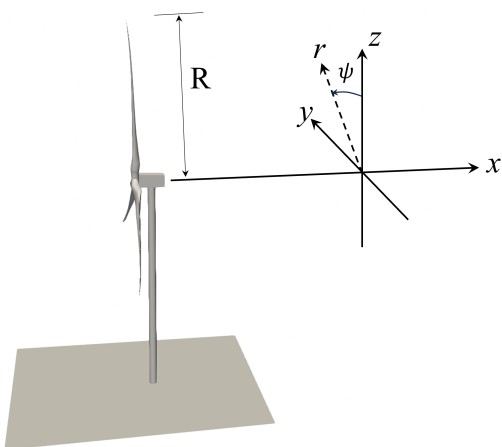

**Figure 2.** Schematic of the wind turbine coordinate system used in this work. The downstream, radial, and azimuthal coordinates are given by $x$, $r$, and $\psi$, respectively, and the turbine rotor radius is given by $R$.

### 2.3 RANS model

In this section, a RANS model is formulated to couple the effects of coherent structures and turbulence on the evolution of the mean velocity field. The model is formulated here in the context of the standard $k - \varepsilon$ RANS closure model (Jones and Launder, 1972), although it may be easily adapted to many turbulence closure modeling approaches. A few assumptions are made to simplify the model, as the focus is on developing a computationally efficient representation of the effects of active wake mixing on the mean flow. First, the dynamics are assumed to be axisymmetric, reducing the complexity of the model to two dimensions. A schematic of the streamwise, radial, and azimuthal coordinates relative to the turbine is shown in figure 2. Second, the boundary layer approximation is applied so that: (1) second order derivatives in the streamwise direction, $x$, are small relative to those in the radial direction, $r$; (2) the radial pressure gradient is decoupled from the velocity field; and (3) turbulent production is dominated by the radial mean streamwise velocity gradient. This leads to a parabolic system that can be marched in the downstream direction, as in Cheung et al. (2024b). Lastly, direct interactions between the coherent structures and the turbulence are neglected so that AWM only forces the evolution of the mean velocity. The resulting equations are

$$\frac{\partial \bar{U}}{\partial x} + \frac{\partial \bar{V}}{\partial r} + \frac{\bar{V}}{r} = 0, \tag{7a}$$

$$\bar{U}\frac{\partial \bar{U}}{\partial x} + \bar{V}\frac{\partial \bar{U}}{\partial r} = \frac{1}{r}\frac{\partial}{\partial r}\left[r(\nu + \nu_t)\frac{\partial \bar{U}}{\partial r}\right] + F_{\mathrm{CS}}, \tag{7b}$$

$$\bar{U}\frac{\partial k}{\partial x} + \bar{V}\frac{\partial k}{\partial r} = \nu_t\left(\frac{\partial \bar{U}}{\partial r}\right)^2 - \varepsilon + \frac{1}{r}\frac{\partial}{\partial r}\left[r(\nu + \nu_t/\sigma_k)\frac{\partial k}{\partial r}\right], \tag{7c}$$

$$\bar{U}\frac{\partial \varepsilon}{\partial x} + \bar{V}\frac{\partial \varepsilon}{\partial r} = \frac{C_{1\varepsilon}\varepsilon}{k}\left[\nu_t\left(\frac{\partial \bar{U}}{\partial r}\right)^2\right] - \frac{C_{2\varepsilon}}{k}\varepsilon^2 + \frac{1}{r}\frac{\partial}{\partial r}\left[r(\nu + \nu_t/\sigma_\varepsilon)\frac{\partial \varepsilon}{\partial r}\right], \tag{7d}$$

where $\bar{U}$ and $\bar{V}$ are the mean streamwise and radial velocity components, respectively. The effects of turbulence, $u'$, on the mean flow are represented by the eddy-viscosity, $\nu_t = C_\mu k^2/\varepsilon$, where $k$ and $\varepsilon$ are the turbulent kinetic energy and dissipation, respectively. To close the $k$-$\varepsilon$ model, the RANS constants, $C_{1\varepsilon}$, $C_{2\varepsilon}$, $C_\mu$, $\sigma_k$, and $\sigma_\varepsilon$, are calibrated based on LES data discussed in Section 2.3.1. The term, $F_{\text{CS}}$, represents forcing of the mean flow by the wave component, $\tilde{\mathbf{u}}$, as

$$F_{\text{CS}} = -\overline{\tilde{u}\frac{\partial \tilde{u}}{\partial x} + \tilde{v}\frac{\partial \tilde{u}}{\partial r}}, \tag{8}$$

and the coupling between the mean component and wave component is discussed further in Section 2.4.

Equations (7a)-(7d) are discretized on a uniform grid in the radial direction using a second-order centered difference method. The radial domain extends to $r_{\max} = 5R$ with a uniform spacing of $\Delta r = 0.025R$. In the $x$-direction, the equations are discretized around the cell centers and a Crank-Nicolson method is used to march $20D$ downstream with uniform step sizes of $\Delta x = 0.1R$. The resulting tridiagonal system is solved using an iterative solver, which advances the solution from one $x$-location to the next.

For each variable, Neumann boundary conditions are applied at $r = 0$,

$$\frac{\partial \bar{U}}{\partial r}(r = 0) = 0, \qquad \frac{\partial k}{\partial r}(r = 0) = 0, \qquad \frac{\partial \varepsilon}{\partial r}(r = 0), \qquad \frac{\partial \bar{V}}{\partial r}(r = 0) = 0,$$

and Dirichlet boundary conditions are applied at $r = r_{\max}$,

$$\bar{U}(r = r_{\max}) = U_\infty, \qquad k(r = r_{\max}) = k_\infty, \qquad \varepsilon(r = r_{\max}) = 0,$$

where $k_\infty/U_\infty^2 = 1.0 \times 10^{-3}$ is specified based on the LES calibration data. Note that the continuity relation (7a) only requires one boundary condition to be imposed on $\bar{V}$.

A hyperbolic tangent profile is used to model the initial condition for $\bar{U}$ at the initial streamwise location, $x = x_0$,

$$\bar{U}(x_0) = 0.5(U_\infty - U_0)\left(1 + \tanh\left(\frac{r - r_e}{\Delta}\right)\right) + U_0, \tag{9}$$

where the nondimensionalized values $U_0/U_\infty = 0.5$, $r_e/R = 1.2$, and $\Delta/R = 0.05$ were determined to provide a good agreement with the azimuthally averaged velocity deficit profiles from the LES data near $x/D = 2$ (see Figures 5 and 6). However, it should be noted that the effects of the nacelle in the near wake are not accounted for in the RANS formulation. The initial profile for $k$ is taken to be proportional to the square of the mean velocity gradient, $k(x_0) \sim (\partial \bar{U}/\partial r)^2 + k_\infty$, such that $\sqrt{3\max(k(x_0))/2} = 0.125$, and the initial $\varepsilon$ is taken to balance turbulent kinetic energy production, i.e., $\varepsilon(x_0) = \sqrt{C_{1\varepsilon}k^2(\partial\bar{U}/\partial r)^2}$.

### 2.3.1 Calibration of RANS

The coefficients, $C_\mu, C_{1\varepsilon}, C_{2\varepsilon}$, of the $k$-$\epsilon$ RANS closure model were calibrated to match the rotor averaged velocities from the baseline LES discussed in 2.5. Since the RANS formulation does not account for the hub and nacelle region from the LES (see fig. 1), the calibration was formulated to match the rotor averaged velocities from a distance of $x/D = 2.0$ to $x/D = 8.0$.

The cost function for this calibration was an $\mathcal{L}_2$ norm error between the RANS output and the LES output. The `L-BFGS-B` (Byrd et al., 1995; Zhu et al., 1997) algorithm as implemented in `scipy` was used for the calibration. The optimal values from this calibration are $C_\mu = 0.0035, C_{1\varepsilon} = 0.163$, and $C_{2\varepsilon} = 2.86$. It is important to note that these values are particular to the initial conditions and RANS closure model used and not a general guideline for wake predictions. It is also to be noted that the calibration is only performed for the baseline cases and not the AWM cases. The constants $\sigma_k$ and $\sigma_\varepsilon$ were not included in the calibration process; instead, the standard values of $\sigma_k = 1.0$ and $\sigma_\varepsilon = 1.3$ proposed by Jones and Launder (1972) were used. Figure 8 shows a close match between the baseline RANS and LES results, showing that these calibrated parameters are representative model constants for capturing the wake behavior in the baseline cases, and will be used for all the RANS results presented in this work.

## 2.4 Linear stability model

In this work, we are primarily interested in evaluating the feasibility of using a wave component model to determine the impact of large-scale coherent structures on the turbine wakes. Many approaches have been used previously in the literature to capture the dynamics of large-scale structures in shear flows, including linear and nonlinear stability analysis (Cheung and Lele, 2009), non-modal stability analysis (Hack and Zaki, 2015), and global stability analysis (Schmid, 2007). These methodologies have been very well developed and successful in analyzing other canonical flows such as pipe flows, boundary layers, and jets.

As an initial step towards demonstrating the feasibility of this modeling approach, a simple parallel flow, inviscid, spatial linear stability analysis was chosen for this work. The focus of the analysis is to model the growth of the large-scale coherent structures and capture resulting changes to the mean flow of the turbine wake with minimal computational effort. Additional effects not captured in this analysis, such as the effects of shear, veer, swirl, or atmospheric stratification, will be included in future analyses.

### 2.4.1 Piecewise constant velocity profile

Analytic solutions to the spatial linear stability problem are possible if we assume the initial wake profile remains axisymmetric and roughly follows a piecewise constant velocity profile. In the current work, we adopt the two-step profile shown in figure 3, which is defined by

$$\overline{U}_{pw}(r) = \begin{cases} U_0, & r < r_1 \\ U_{\text{half}}, & r_1 \leq r \leq r_2 \\ U_\infty, & r > r_2 \end{cases} \tag{10}$$

where $U_0$ is the centerline velocity, $U_\infty$ is the freestream velocity, and $U_{\text{half}} = \frac{1}{2}(U_0 + U_\infty)$ is the averaged velocity of the wake shear region from $r_1 \leq r \leq r_2$. During the solution process, the $r_1$ and $r_2$ parameters can be chosen so that the displacement, $\delta$, and momentum, $\delta_\theta$, areas of the $\overline{U}_{pw}$ profile match that same displacement and momentum areas calculated from the RANS mean flow profiles, $\overline{U}_{RANS}$. Using the following definitions for $\delta$ and $\delta_\theta$:

$$\delta(\overline{U}) = 2\pi \int_0^\infty \left(1 - \frac{\bar{U}(r)}{U_\infty}\right) r \, dr, \tag{11}$$

$$\delta_\theta(\overline{U}) = 2\pi \int_0^\infty \frac{\bar{U}(r)}{U_\infty} \left(1 - \frac{\bar{U}(r)}{U_\infty}\right) r \, dr, \tag{12}$$

where $r_1$ and $r_2$ are then found by solving the following algebraic system:

$$\delta(\overline{U}_{pw}) = \delta(\overline{U}_{RANS}), \tag{13a}$$

$$\delta_\theta(\overline{U}_{pw}) = \delta_\theta(\overline{U}_{RANS}). \tag{13b}$$

An additional simplification is possible if we assume that the wake shear region remains small relative to the size of the rotor diameter. In this case, we can decompose the $\overline{U}_{pw}$ profile into

$$\overline{U}_{pw}(r) = \overline{U}^{(0)}(r) + \overline{U}^{(1)}(r), \tag{14}$$

where $\overline{U}^{(0)}$ is the Heaviside step function,

$$\overline{U}^{(0)}(r) = \begin{cases} U_0, & r \leq r_e \\ U_\infty, & r > r_e, \end{cases} \tag{15}$$

and $\overline{U}^{(1)}$ is a small perturbation to the single step profile:

$$\overline{U}^{(1)}(r) = \begin{cases} 0, & r < r_1 \\ +\Delta U, & r_1 \leq r \leq r_e \\ -\Delta U, & r_e \leq r \leq r_2 \\ 0, & r > r_2. \end{cases} \tag{16}$$

This assumption allows the analytical results of Batchelor and Gill (1962) to be directly applied with some minor modifications, as discussed in the following section.

### 2.4.2 Spatial linear stability formulation

For the flow variables $\tilde{\phi} = [\tilde{u}, \tilde{v}, \tilde{w}, \tilde{p}]$, where $\tilde{u}$, $\tilde{v}$, and $\tilde{w}$ are the streamwise, radial, and azimuthal velocities, respectively, and $\tilde{p}$ is pressure, we assume that they can be expressed in terms of the radial eigenfunctions, $\hat{\phi}_n$, and the complex exponential basis functions

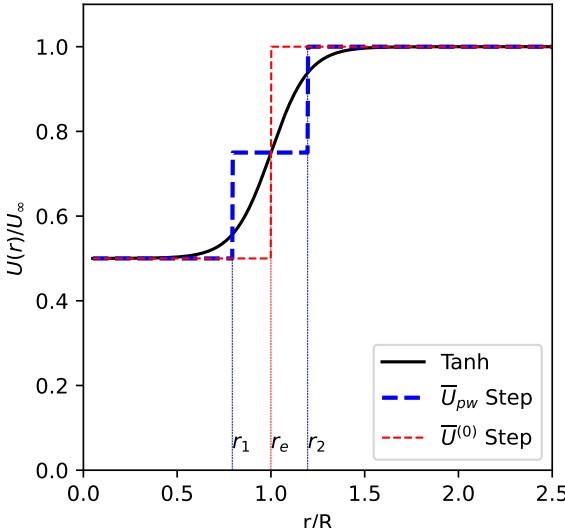

**Figure 3.** Comparison of the step profile with the hyperbolic tangent profile

$\tilde{\phi}(x, r, \psi, t) = \hat{\phi}_n(r) e^{i\alpha x + in\psi - i\omega t}$,                                                       (17)

where $\alpha = \alpha_r + i\alpha_i$ is the complex streamwise wavenumber, $n$ is the azimuthal mode number, and $\omega$ is the temporal frequency. Here the value of $n$ and $\omega$ are taken to match the azimuthal mode number and Strouhal frequencies used in the desired blade pitching strategy from equation (1) and described in section 2.5.1 and table 3. In this study, we consider the impact of a single instability wave, at a single Strouhal number and a specific $n$ on the turbine wake development. For the more general case,

multiple instability wave components can be included in the analysis, and a summation over all wave components is required in equation (17).

Assuming an inviscid, parallel flow with a piecewise constant velocity profile, the governing equations for mass and momentum conservation of the wave components, $\tilde{\phi}$, are

$$\frac{\partial \tilde{u}}{\partial x} + \frac{\partial \tilde{v}}{\partial r} + \frac{\tilde{v}}{r} + \frac{1}{r}\frac{\partial \tilde{w}}{\partial \psi} = 0, \tag{18a}$$

$$\frac{\partial \tilde{u}}{\partial t} + \overline{U}\frac{\partial \tilde{u}}{\partial x} = -\frac{1}{\rho}\frac{\partial \tilde{p}}{\partial x}, \tag{18b}$$

$$\frac{\partial \tilde{v}}{\partial t} + \overline{U}\frac{\partial \tilde{v}}{\partial x} = -\frac{1}{\rho}\frac{\partial \tilde{p}}{\partial r}, \tag{18c}$$

$$255 \quad \frac{\partial \tilde{w}}{\partial t} + \overline{U} \frac{\partial \tilde{w}}{\partial x} = -\frac{1}{\rho r} \frac{\partial \tilde{p}}{\partial \psi}. \tag{18d}$$

Here, $\overline{U}(r) = \overline{U}_{pw}$, with the parameters $r_1$, $r_2$, $U_0$, and $U_\infty$ chosen to match the characteristics of the RANS wake profiles. Inserting the representation (17) in equations (18) leads to the following spectral versions of the governing equations

$$i\alpha \hat{u}_n + \frac{\tilde{v}_n}{r} + \frac{\partial \tilde{v}_n}{\partial r} + \frac{in}{r} \tilde{w}_n = 0, \tag{19a}$$

$$260 \quad \xi \hat{u}_n + \hat{v}_n \frac{d\overline{U}}{dr} + \hat{f}_n^{(1)} = -i\frac{\alpha}{\rho} \hat{p}_n, \tag{19b}$$

$$\xi \hat{v}_n = -\frac{1}{\rho} \frac{d\hat{p}_n}{dr}, \tag{19c}$$

$$\xi \hat{w}_n = -i\frac{n}{\rho r} \hat{p}_n, \tag{19d}$$

where $\xi(r) = \alpha \overline{U}^{(0)}(r) - \omega$. Note that the $\overline{U}_{pw}$ profile has been decomposed according to equation (14) and the term $\hat{f}_n^{(1)} = i\alpha \overline{U}^{(1)} \hat{u}_n$. Equations (19) can be combined into the Rayleigh ordinary differential equation for the pressure, $\hat{p}_n$,

$$\frac{1}{r} \frac{d}{dr} \left( r \frac{d\hat{p}_n}{dr} \right) - \left[ \left( \frac{n}{r} \right)^2 + \alpha^2 \right] \hat{p}_n = i\rho \alpha \hat{f}_n^{(1)}. \tag{20}$$

Both the eigenfunctions, $\hat{p}(r)$, and the eigenvalues, $\alpha$, can be decomposed into a zeroth order and first order components

$$\hat{p}_n = \hat{p}_n^{(0)} + \hat{p}_n^{(1)}, \tag{21a}$$

$$270$$

$$\alpha = \alpha^{(0)} + \alpha^{(1)}. \tag{21b}$$

Here, both $\hat{p}_n^{(1)}$ and $\alpha^{(1)}$ are assumed to be small relative to $\hat{p}_n^{(0)}$ and $\alpha^{(0)}$, respectively, and the solution can be found as part of an eigenvalue perturbation problem. Equation (20) can be similarly divided into the zeroth order and first order contributions, where only the appropriate order terms are included in the equations:

$$275 \quad \mathcal{L}\{\hat{p}_n^{(0)}\} = \frac{1}{r} \frac{d}{dr} \left( r \frac{d\hat{p}_n^{(0)}}{dr} \right) - \left[ \left( \frac{n}{r} \right)^2 + \left( \alpha^{(0)} \right)^2 \right] \hat{p}_n^{(0)} = 0, \tag{22a}$$

$$\mathcal{L}\{\hat{p}_n^{(1)}\} + 2\alpha^{(0)} \alpha^{(1)} \hat{p}^{(0)} = i\rho \alpha^{(0)} \hat{f}_n^{(1)}. \tag{22b}$$

The zeroth order solution to equation (22a) is given by the modified Bessel functions

$$
\hat{p}_n^{(0)}(r) = \begin{cases} C_1 I_n(\alpha^{(0)}r), & r < r_e \\ C_2 K_n(\alpha^{(0)}r), & r \geq r_e. \end{cases}
\tag{23}
$$

The constants, $C_1$ and $C_2$, are chosen so the pressure is continuous at $r = r_e$, and the kinematic condition for the displacement, $\eta$, of a material line at $r = r_e$ is also satisfied:

$$
\frac{\partial \tilde{\eta}}{\partial t} + \overline{U}\frac{\partial \tilde{\eta}}{\partial x} = \tilde{v}.
\tag{24}
$$

Assuming the functional form $\tilde{\eta}(x, \psi, t) = \hat{\eta}e^{i\alpha x + in\psi - i\omega t}$, enforcing above conditions leads to the following nonlinear relation which can be used to solve for $\alpha^{(0)}$ at every frequency $\omega$:

$$
\frac{\xi(r_0)^2}{\xi(r_\infty)^2} = \frac{K_n'(\alpha^{(0)}r_e)I_n(\alpha^{(0)}r_e)}{K_n(\alpha^{(0)}r_e)I_n'(\alpha^{(0)}r_e)}.
\tag{25}
$$

Up to this point, the analysis follows that of Batchelor and Gill (1962) for piecewise constant velocities and is shown to be valid for infinitely sharp, top-hat velocity profiles. However, in the current work, capturing the effects of the wake spreading are important to the growth and evolution of large-scale structures. This can be accomplished by including a small perturbation to the $\overline{U}^{(0)}$ profile and calculating the corresponding perturbation to the growth rates. Once $\hat{p}_n^{(0)}$ and $\alpha^{(0)}$ are known, the perturbation, $\alpha^{(1)}$, to the wavenumber can be found by applying the inner product,

$$
\langle f, g \rangle = \int_0^\infty f(r)g(r) \, r \, dr,
\tag{26}
$$

to equation (22b), leading to

$$
\langle \mathcal{L}\{\hat{p}_n^{(1)}\}, \hat{p}_n^{(0)} \rangle + \langle 2\alpha^{(0)}\alpha^{(1)}\hat{p}^{(0)}, \hat{p}_n^{(0)} \rangle = \langle i\rho\alpha^{(0)}\hat{f}_n^{(1)}, \hat{p}_n^{(0)} \rangle.
\tag{27}
$$

Because $\hat{p}_n^{(0)}$ is self-adjoint and satisfies equation (22a), the terms in equation (27) can be rearranged into the following expression for $\alpha^{(1)}$:

$$
\alpha^{(1)} = -\frac{C_\alpha(\alpha^{(0)})^2 \int_0^\infty \overline{U}^{(1)}\dfrac{\hat{p}_n^{(0)}\hat{p}_n^{(0)}}{\alpha^{(0)}\overline{U}^{(0)} - \omega} \, r dr}{2\int_0^\infty \hat{p}_n^{(0)}\hat{p}_n^{(0)} r \, dr}
\tag{28}
$$

where $C_\alpha$ is complex calibration constant. The full eigenvalue, $\alpha$, can then be reconstructed through equation (21b). The real part of the wavenumber, $\alpha_r$, determines the streamwise wavelength of the large-scale coherent structures, while the imaginary component, $\alpha_i$, dictates the spatial growth of the structures.

To examine the accuracy of this asymptotic, analytic approach with a piecewise constant velocity profile, a comparison of the linear stability solution using a continuous hyperbolic tangent profile (16) and the $\overline{U}_{pw}$ profile from (10) is shown in figure 4.

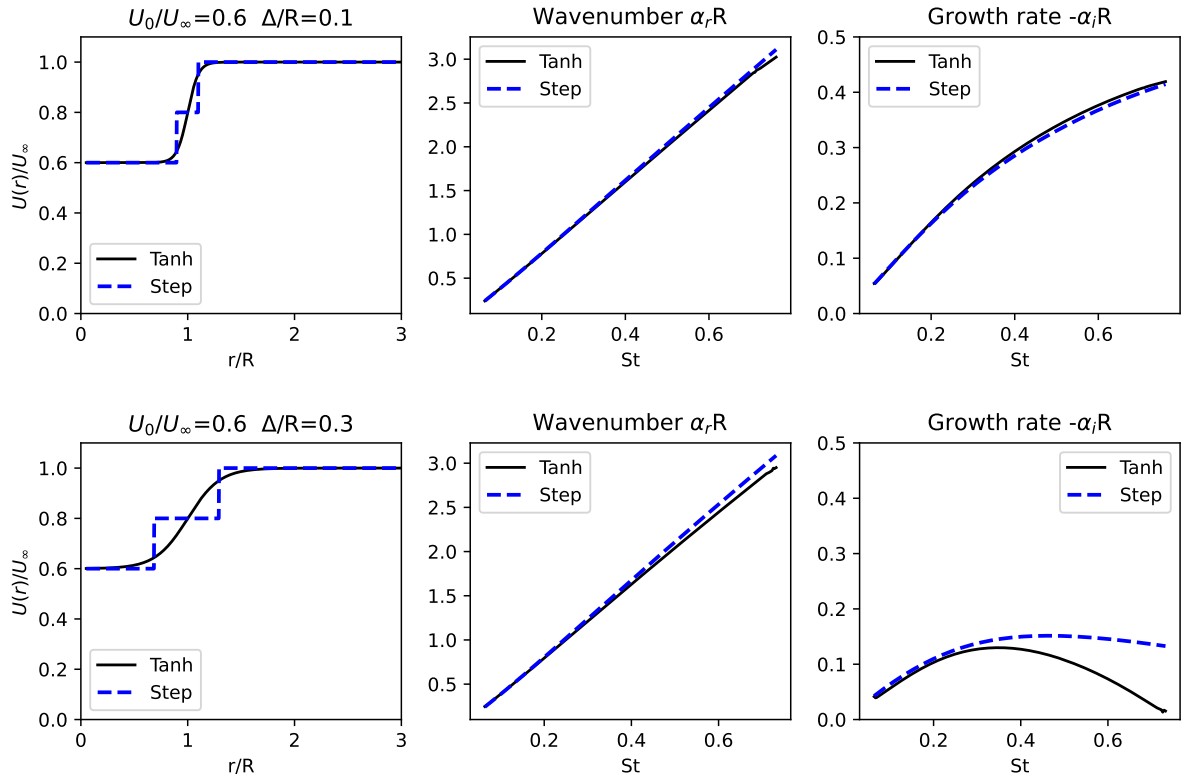

**Figure 4.** Comparison of linear stability theory between the analytic tanh profiles (16) and the piecewise step profiles (10). The first helical mode, $n = 1$, is shown in all cases.

As expected, the dispersion relation, $\alpha = \alpha(\omega)$, shows excellent agreement between the approaches when the profile width, $\Delta$, is relatively small. For larger values of $\Delta$, the wavenumber, $\alpha_r$, calculated using the piecewise constant profile approximation $U_{pw}$, still agreed with the hyperbolic tangent profile, although there were some discrepancies visible for the growth rate, $\alpha_i$. However, for the lower frequencies of interest, the agreement between the approaches is still reasonably accurate.

### 2.4.3 Coupled solution process

The spatial linear stability formulation described in section 2.4.2 can be easily integrated into the RANS solution process discussed in section 2.3. At every streamwise location, $x$, the RANS velocity profile is first computed assuming $F_{CS} = 0$. This velocity profile is then used in the linear stability formulation to compute the velocity eigenfunctions, $\hat{u}$ and $\hat{v}$, and the corresponding wavenumbers, $\alpha$. The evolution of the wave component of the flow variables can be calculated using the formula

$$\tilde{u}(x, r, \psi, t) = a_n \hat{u}_n(r) \exp\left( i \int \alpha(x) \, \mathrm{d}x - in\psi - i\omega t \right), \tag{29}$$

where the integral over $\alpha$ in the exponential accounts for the slow changes in growth rate as the mean flow evolves. The initial amplitude of the wave component is given by $a_n$. Once $\tilde{u}$ and $\tilde{v}$ are known, the mean flow correction term, $F_{CS}$, can be calculated, and a new RANS velocity profile can be computed for the same $x$ location. This process is repeated until the RANS velocity profiles meet a specified convergence criteria (for which we use the Frobenius norm of the difference between two successive solutions being less than $10^{-7}$), after which the streamwise marching process proceeds to the next location at $x + \Delta x$. This process results in a two-way coupled model of the mean-flow and the coherent flow structures, which differs from other dynamic approaches, such as the Dynamic Wake Meandering model, that use a pre-determined mean-flow to drive wake dynamics.

The initial formulation of both the RANS model and the linear stability model was implemented in Python and run on workstations with a single CPU for all cases. For typical cases, which used 200 grid points in the radial direction and 200 streamwise points, the baseline RANS calculation took 1-2 s to compute, and in cases with the RANS model coupled to the linear stability model, the total solve time was approximately 11-12 s.

## 2.5 AMR-Wind LES calculations

To generate the data necessary to calibrate the RANS model coefficients and evaluate the accuracy of the coupled RANS and linear stability approach, a series of LES of turbine wakes was performed. These were done with the AMR-Wind code (Sharma et al., 2024; Sprague et al., 2020; Kuhn et al., 2025), a massively parallel, block-structured adaptive-mesh, incompressible flow solver for wind turbine and wind farm simulations. AMR-Wind solves the incompressible and low Mach formulations of the Navier-Stokes equations, as well as temperature, subgrid-scale kinetic energy, and other scalar equations necessary for large eddy simulation (LES) of wind farms. AMR-Wind solves the discretized equations using a second order finite method and second order temporal integration. AMR-Wind includes all the necessary physics modules to simulate atmospheric boundary layers (ABLs). Included in this effort are ABL forcing, Boussinesq buoyancy, Coriolis forcing, body forcing to maintain the precursor-derived inflow condition in the presence of the turbine's blockage, and body forcing from coupling to OpenFAST (Jonkman et al., 2018; NREL, 2023) for turbine representation using actuator line models (these are the same forcing terms used in Brown et al. (2025) and Hsieh et al. (2025), for instance). AMR-Wind leverages AMReX for data structures, parallelism abstractions, and performance portability on heterogeneous architectures (Zhang et al., 2019). This framework has shown computational performance across a wide range of systems and applications (Fedeli et al., 2022; Henry de Frahan et al., 2022, 2024).

### 2.5.1 Turbine simulation parameters

Simulations using the IEA 15 MW reference turbine and the atmospheric conditions listed in section 2.1 were performed in AMR-Wind using the one equation $k_{sgs}$ LES model (Moeng, 1984) and an actuator line model coupled to OpenFAST to represent the turbine blade forces. The simulation domains were either 4.5 km $\times$ 2 km $\times$ 1 km (Med WS case), or 6.7 km $\times$ 2 km $\times$ 1 km (Low WS and High WS cases). In all cases, a background mesh resolution of 5 m was used, which was refined to a resolution of 2.5 m in a region 4.75D upstream to 12D downstream of the rotor, leading to mesh sizes of 179M and 309M,

**Table 3.** AWM parameters

| Name | Mode ($n$) | Amplitude ($A$) | Strouhal number ($St$) | Clocking Angle ($\psi_{\text{clock}}$) |
|---|---|---|---|---|
| Baseline | N/A | N/A | N/A | N/A |
| HelixA2 | -1 | 2 deg | 0.30 | 90 deg |
| HelixA4 | -1 | 4 deg | 0.30 | 90 deg |
| PulseA2 | 0 | 2 deg | 0.30 | 90 deg |
| PulseA4 | 0 | 4 deg | 0.30 | 90 deg |

respectively. A timestep of 0.02 s was used in the turbine simulations, and all simulations had a total runtime of at least 1000 s to allow the initial transients to dissipate and the wake structures to fully develop in the flow.

OpenFAST is a conglomeration of models that characterize the whole-system dynamics of a wind turbine including the aerodynamic and structural loading, drive train, and controls. Each of these individual models were tuned to match the specifics of the IEA 15 MW turbine. In addition, there are various submodels and parameters within the primary OpenFAST modules that can be adjusted to affect the fidelity, discretization, and properties of the simulations. For example, the aerodynamics module in OpenFAST consists of six submodels for rotor/wake induction, blade airfoil aerodynamics, tower influence on the blade, tower and nacelle drag, aeroacoustics, and buoyancy effects for floating substructure components. For OpenFAST, there were no wake, induction, hub-loss, or tip-loss models enabled as those phenomena were modeled within AMR-Wind. The blade airfoil structural dynamics were calculated through ElastoDyn and the unsteady aerodynamics were represented by the Beddoes-Leishman unsteady model with the Minnema/Pierce extension. Blade definition files are provided to the aerodynamics module to define the blade nodal discretization, geometry, chord, twist, airfoil identifiers, and buoyancy properties. Fifty and twenty nodes were used to represent the aerodynamics on the turbine blades and the tower, respectively. Similarly, the aerodynamic effects of the hub and nacelle were represented using actuator forcing with a drag coefficient of 0.5 and a representative area of 49.5 m$^2$. Precise subsampling by OpenFAST was performed with a subcyling ratio between OpenFAST and AMR-Wind of 4:1.

### 2.6 Blade pitch actuation for AWM

The earlier work of (Cheung et al., 2024a) explored the connection between blade pitch actuation parameters used in AWM strategies and the instability modes considered in the current analysis. In that study, different blade pitch actuation strategies were applied to an OpenFAST turbine model simulation using different pitch amplitudes, azimuthal mode numbers, and the desired Strouhal frequency of St=0.30. An analysis of the resulting blade loads showed that there was a corresponding fluctuating streamwise blade force that appeared at the same azimuthal mode number and Strouhal frequency. Through a spectral POD analysis, we can see that these fluctuating streamwise blade forces then excite a similar response in the near wake (Yalla et al., 2025). Although there may be differences in the radial profiles between the streamwise forces induced by fluctuating blade motions and the eigenfunctions of the Rayleigh equation (22), it is sufficient to pitch the blades at the specified azimuthal mode number $n$ and Strouhal number to excite the desired instability mode.

In the baseline turbine simulations, no AWM strategy was employed and the wake was allowed to develop naturally without any blade pitch actuation. These baseline cases were compared to simulations in which the helix and pulse AWM strategies were used (table 3). All AWM strategies used a single actuation frequency of St $= 0.30$, which is consistent with the Strouhal forcing used in previous studies (Cheung et al., 2024a). The blade pitch amplitudes were set to either 2 degrees or 4 degrees in both the helix and the pulse AWM strategies to determine the relative effectiveness of each actuation strategy.

## 3 Results

To evaluate the accuracy of the RANS and linear stability wake model, we compare the modeled wake behavior with the corresponding wake behavior from the AMR-Wind simulations. Results are shown first for the baseline cases where no AWM strategy was used, which allows us to evaluate the underlying RANS model without any coupling to the linear stability model. This is followed by a discussion of the AWM cases with helix and pulse actuation and an evaluation of the full RANS plus linear stability model.

### 3.1 Baseline wake behavior

Comparison of the hub-height velocity profiles between the RANS model and the AMR-Wind LES calculation for the Med WS and High WS cases are shown in figures 5 and 6, respectively, for various downstream distances. The baseline wake behavior for the Low WS case was very similar to the Med WS case because the turbine was operating at the same thrust coefficient, so the Low WS comparisons are not shown below for the sake of brevity.

In the medium to far wake regions, for downstream distances of $x/D > 3.0$, good agreement is seen between the wake profiles from the RANS model and the AMR-Wind calculations. The general evolution of the wake deficit and the wake spreading behavior is well captured by the parabolized RANS model. The AMR-Wind wake profiles show evidence of veer effects, which causes asymmetry in the LES wake profiles. This effect is not captured by the RANS model due to the axisymmetric formulation, but the overall match between the methods remains high.

Very close to the turbine rotor some differences between the wake profiles are noticeable. For streamwise distances of $x/D < 3.0$, we see the influence of the hub and nacelle on LES wake profiles, which is not captured in the RANS model. The actuator line representation of the turbine in AMR-Wind more accurately models the aerodynamics near the hub and root sections of the blades, leading to a small recirculation zone immediately downstream of the nacelle. The simplified nature of the initial RANS profiles neglects these effects, as well as any asymmetries due to the interactions of swirl with shear and veer in addition to speedup of the ambient flow from wake blockage. However, despite these approximations, the RANS model still accurately captures the velocity shear near the wake edges.

A comparison of the centerline and rotor averaged velocities, shown in figures 7 and 8, provides a similar picture of the RANS model's accuracy for the baseline wake cases. In the far wake region, the RANS model accurately predicts the recovery of the centerline and rotor averaged velocities. Very close to the rotor, the RANS model assumes the presence of a potential core region in the wake, which is not realistic, so it is unsurprising that the centerline velocities do not agree until $x/D \approx 4.0$.

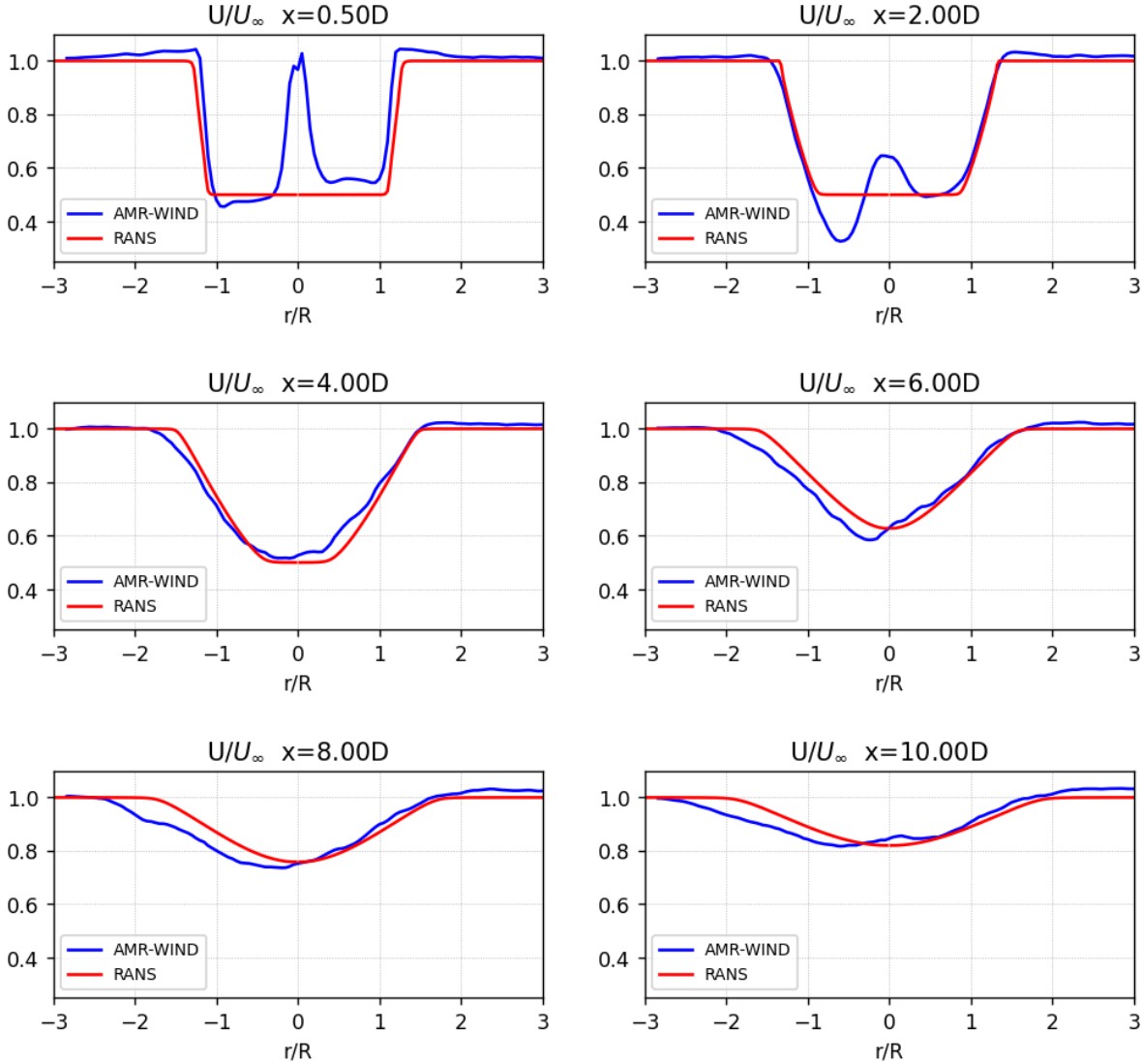

**Figure 5.** Comparisons of the normalized hub-height velocity profiles from the RANS model and AMR-Wind calculations for the baseline Med WS case.

Some differences are observed in the RANS behavior between the High WS and the Low/Med WS cases. One noticeable difference is that the potential core region is correctly modeled in the Low and Medium WS cases but overestimated in the High WS case. This is reflected in the comparisons of figures 7 and 8, as well as the wake recovery in the velocity profiles figure 6, which show lower centerline and rotor averaged velocities for the RANS High WS case in the far wake. We believe that these discrepancies can be reduced through improvements in the RANS model and additional calibration across a wider variety of wake cases in future work. It is acknowledged that a comparison to an axisymmetric LES of just the turbine rotor would

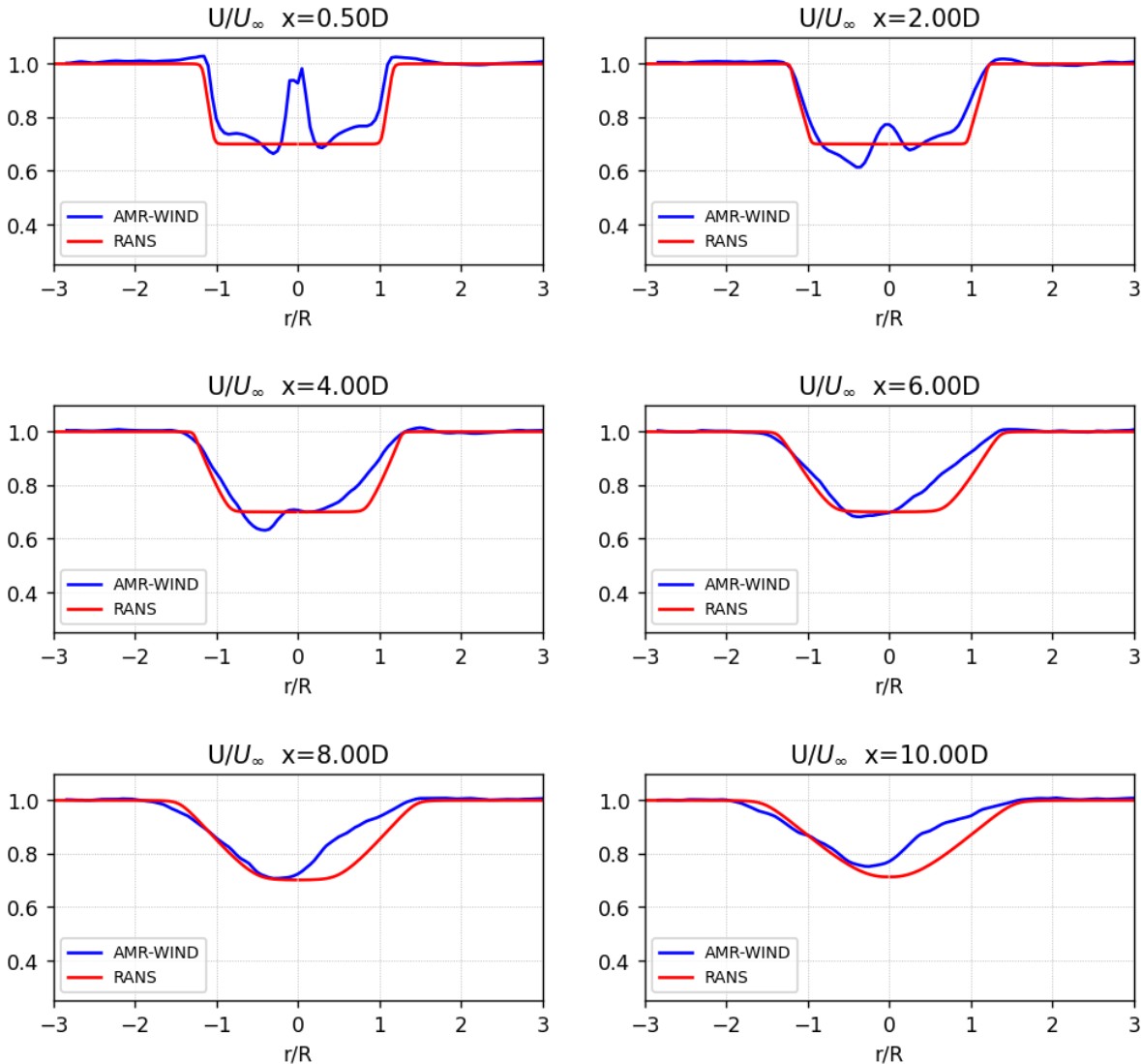

**Figure 6.** Comparisons of the normalized hub-height velocity profiles from the RANS model and AMR-Wind calculations for the baseline High WS case.

have yielded a direct comparison with RANS. However, the objective of this work is to provide a usable, proof-of-concept framework that illustrates how a RANS model with a linear stability model can capture most of the phenomena of interest in the LES data. The long-term goal is to build it up from common principles towards being able to capture increasing physics complexity, such as shear effects, veer, and asymmetry. The differences between the RANS and LES discussed here and in

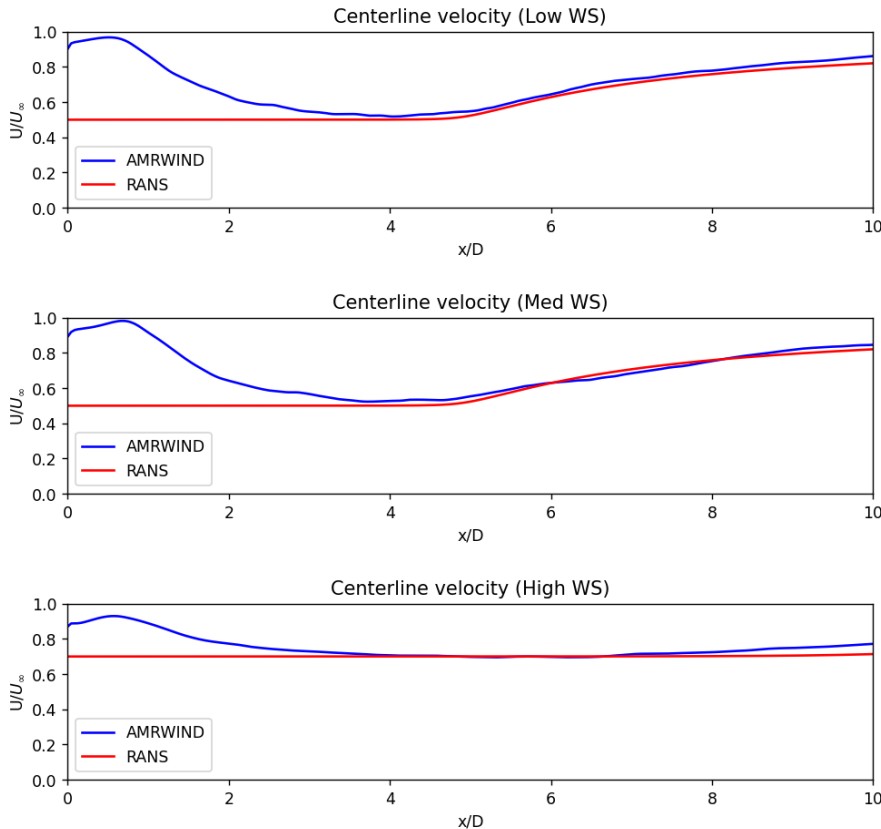

**Figure 7.** Comparison of the normalized centerline velocities for the baseline turbine wakes.

other sections of the paper highlight that the current approach performs well in comparison with complex LES data while also pointing towards future improvements.

### 3.2 Comparisons for AWM cases

With the application of an AWM strategy, we expect the turbine wake to mix faster due to the presence of the large-scale coherent structures. As shown in figures 9 and 10, the hub-height velocity profiles for the LES calculations and the RANS with linear stability model indicate a faster wake recovery and increased mixing in the downstream wake. For the Med WS case in figure 9, there was qualitative agreement between the LES calculations and RANS with linear stability model in predicting the changes to the wake width and centerline velocity for both the helix and pulse AWM cases and for both $2°$ and $4°$ pitch actuation. For the helix AWM case at the High WS condition (figure 10), the LES calculations show more impact to the centerline velocity recovery, although the RANS with linear stability model still shows the changes to the wake width due to AWM.

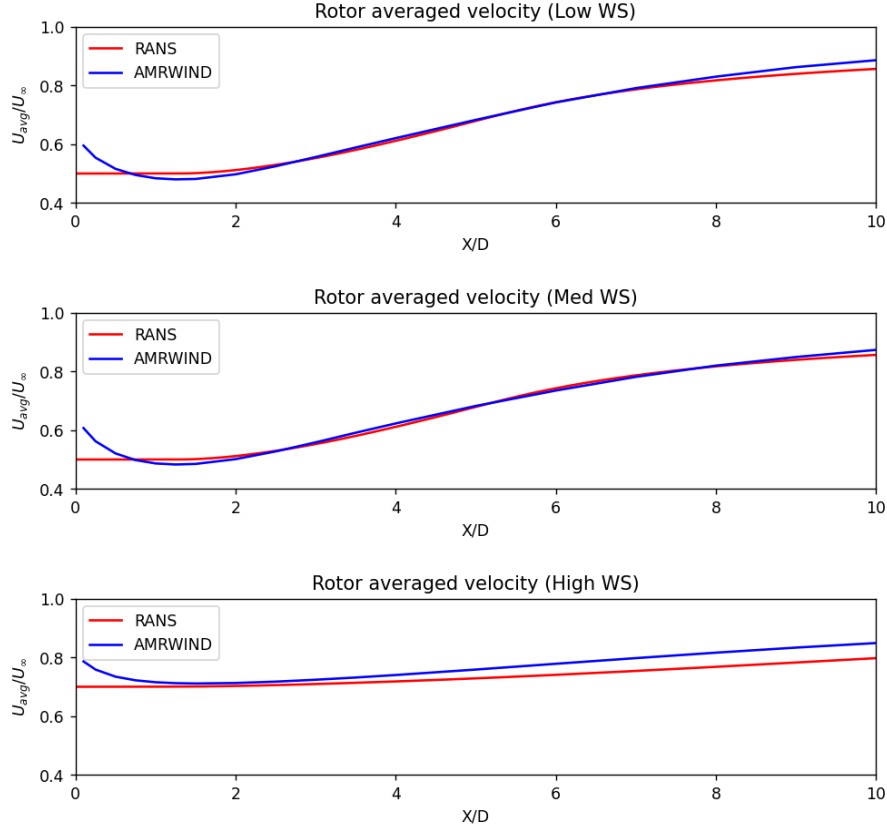

**Figure 8.** Comparison of the normalized rotor averaged velocities for the baseline turbine wakes.

The rotor averaged velocity comparisons in figures 11 and 12 show a similar level of agreement between the AMR-Wind and RANS with linear stability model in the far wake. For downstream distances of $x/D > 5.0$, the RANS with linear stability model qualitatively captured the wake recovery benefits for both the helix and pulse approaches. It was also seen that the helix AWM was not as effective as in the Med WS case as it was in the High WS condition. This is attributed to the fact that the turbine operates at a lower thrust coefficient at the higher wind speeds, resulting in less initial wake deficit and lower velocity shear near the wake edges. The lower shear in the turbine wake translates to slower growth of the large-scale coherent structures, meaning that there is less opportunity for them to mix the turbine wake and impact the flow.

Some differences between the LES calculations and the RANS with linear stability model are observed in the near wake region. For the Low WS and the Med WS cases, the growth of the coherent structures in the LES calculations is faster than the RANS with linear stability model, so the wake benefits to the rotor averaged velocity also appear earlier in the flow. However, in both the LES and the RANS with linear stability model, the growth of the large-scale structures saturate at similar levels downstream, so the final wake benefits in the far wake remain comparable.

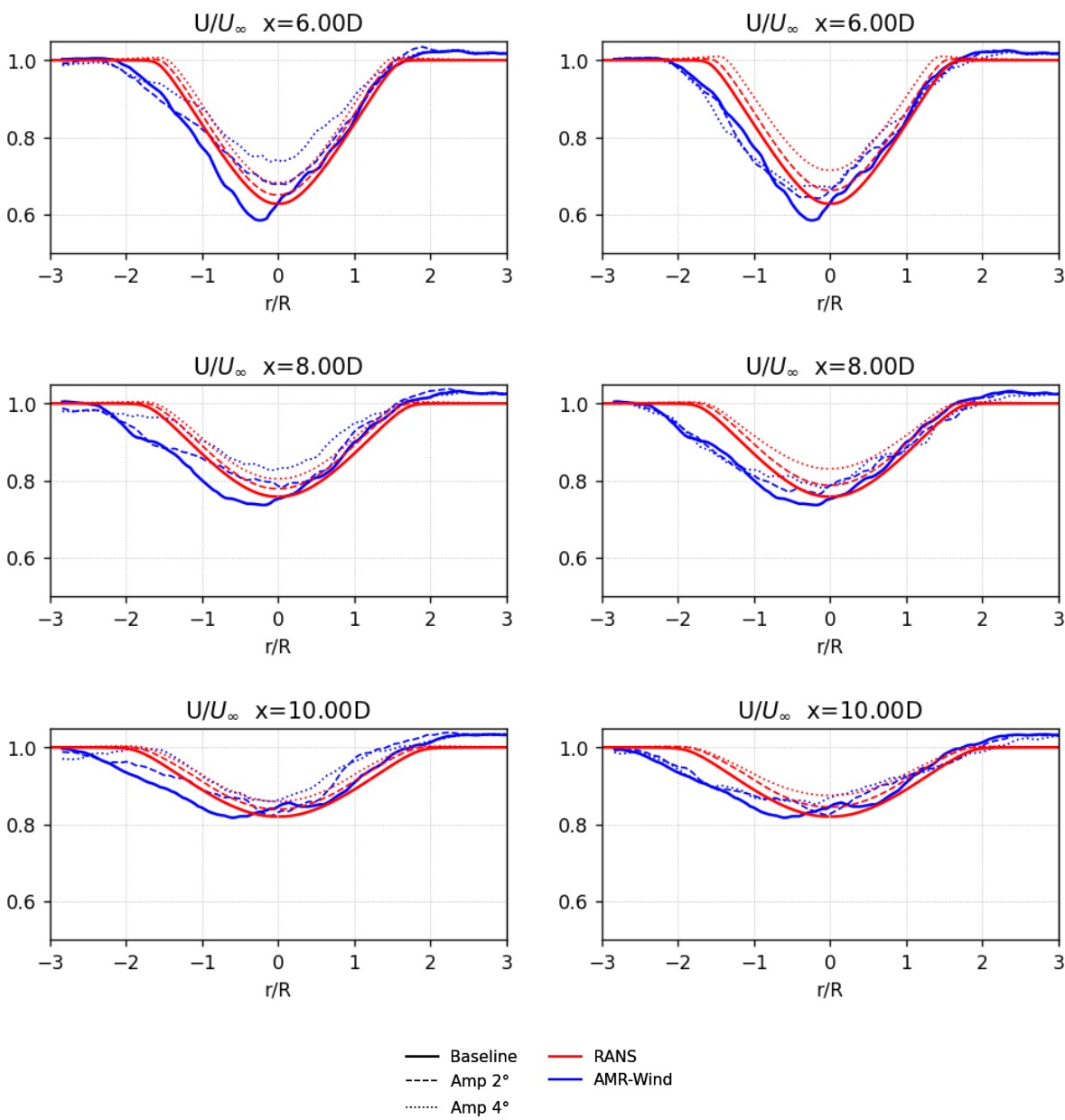

**Figure 9.** Comparisons of the hub-height velocity profiles from AMR-Wind and RANS with linear stability model for the Med WS case with Helix AWM (right column) and Pulse AWM (left column) actuated.

Note that the observed AWM model behavior for the High WS case is consistent with earlier observations regarding the RANS model predictions for that condition. In figure 10, the larger potential core region in the RANS profiles limits the modifications from the coherent structures to the wake shear regions until father downstream in the wake. This leads to relatively

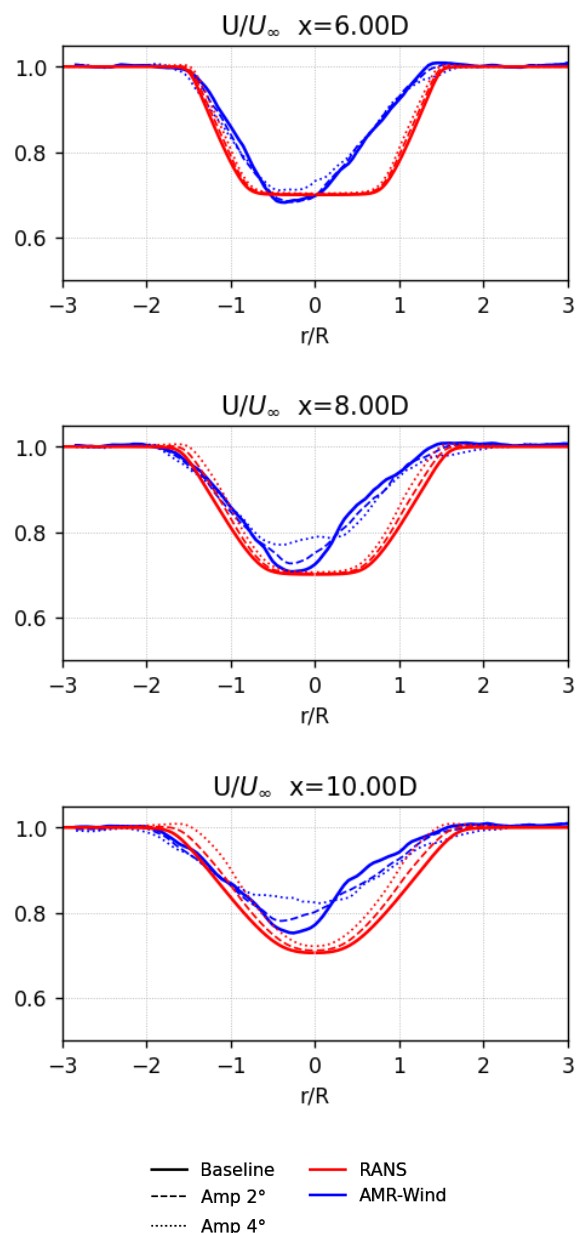

**Figure 10.** Comparisons of the hub-height velocity profiles from AMR-Wind and RANS with linear stability model for the High WS case with Helix AWM.

minor changes to the centerline velocity for the High WS case compared to the Med or Low WS cases (figure 9) and suggests that accurately capturing the mean flow is critical to modeling the impact of large-scale structures on wake behavior.

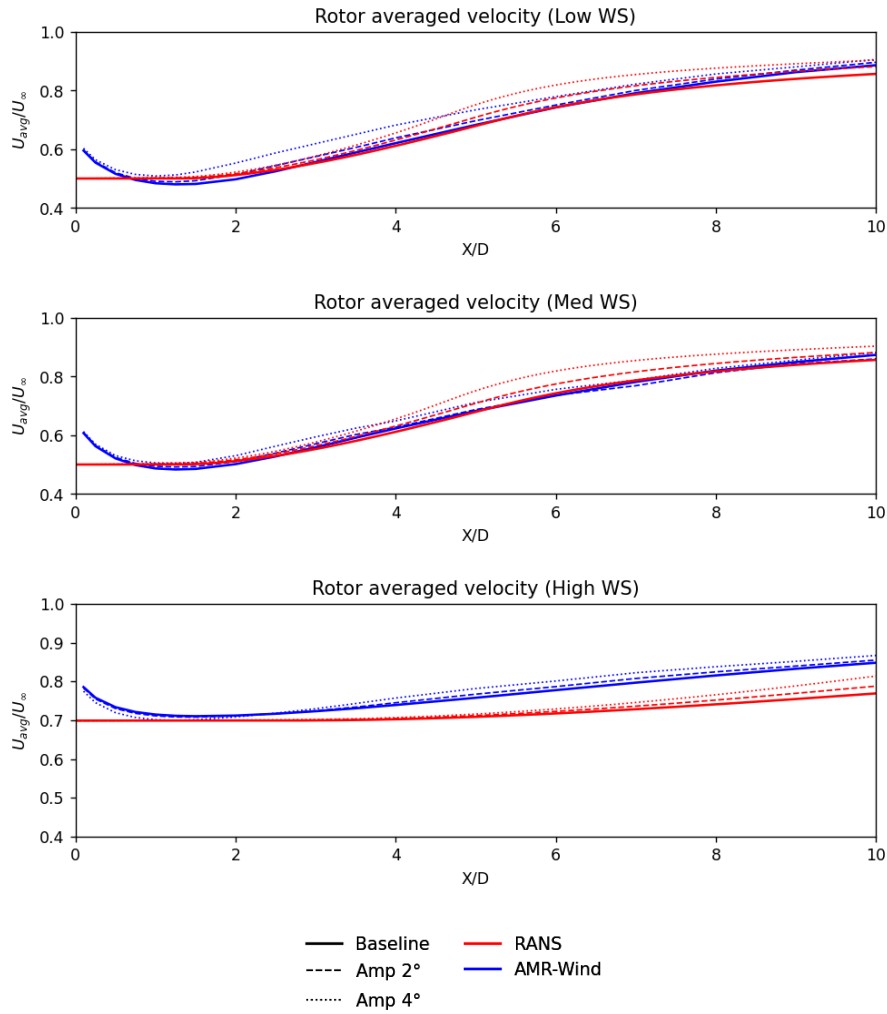

**Figure 11.** Comparisons of the normalized rotor averaged velocities from the Low WS, Med WS, and High WS cases using the helix AWM strategy with $2°$ and $4°$ actuation amplitudes.

A quantitative measure for the accuracy of the RANS and linear stability approach to modeling AWM effects is provided in table 4. Comparisons of the minimum hub-height streamwise velocities between the AMR-Wind calculations and the modeled wake predictions are shown at $x/D = 8$ and $x/D = 9$. For the Med and Low WS cases with helix and pulse forcing, the majority of the velocity errors are below 5%, and, as expected, the largest differences compared to AMR-Wind occurred for the High WS cases.

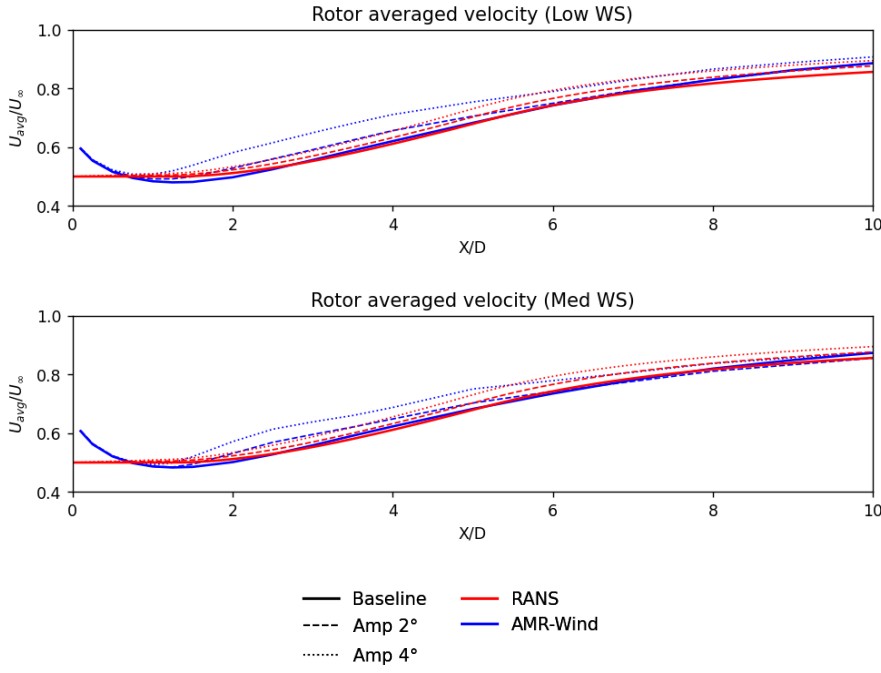

**Figure 12.** Comparisons of the normalized rotor averaged velocities from the Low WS and Med WS cases using the pulse AWM strategy with 2° and 4° actuation amplitudes.

### 3.2.1 Phase averaged velocity

Some insight to the behavior of the large-scale structures can be seen in figure 13, where contours of the mean and phase averaged velocity fields is shown for the Low WS case. As expected, the near wake region of the RANS and linear stability model fails to capture some flow features immediately downstream of the rotor disk. However, in the far wake region, the differences in the coherent structures between the pulse AWM forcing and the helix AWM forcing become apparent. The wave components in the pulse case form axisymmetric structures with a wavelength of approximately 2D, while a spiral pattern appears in the helix AWM cases with similar wavelength. In the AMR-Wind simulations the structures emerge earlier in the turbine wake, but eventually saturate and impact the wake in a qualitatively similar way compared to the RANS and linear stability model predictions.

### 4 Conclusions

In this work, a framework for modeling AWM was developed that accounts for the effects of both the large-scale coherent structures and the turbulence on the mean flow. Using a triple-decomposition approach, the turbine wake flow was separated into a time-averaged mean flow, fine scale turbulent, and phase averaged components, and a computationally efficient method

**Table 4.** Comparisons of the minimum hub-height streamwise velocity, $\min(U)/U_\infty$, between the AMR-Wind calculations and the RANS with linear stability model at $x/D = 8$ and $x/D = 10$. The error is defined as $\epsilon = U_{RANS}/U_{AMRWIND} - 1$.

(a) Low WS pulse and helix methods

|  | $x/D = 8$ | | | $x/D = 10$ | | |
|---|---|---|---|---|---|---|
|  | Baseline | $A = 2°$ | $A = 4°$ | Baseline | $A = 2°$ | $A = 4°$ |
| AMR-Wind (Pulse) | 0.778 | 0.793 | 0.845 | 0.856 | 0.868 | 0.899 |
| RANS+LST (Pulse) | 0.759 | 0.779 | 0.805 | 0.820 | 0.839 | 0.860 |
| Error [%] | -2.4 | -1.7 | -4.8 | -4.2 | -3.3 | -4.4 |
| AMR-Wind (Helix) | 0.778 | 0.774 | 0.812 | 0.856 | 0.868 | 0.866 |
| RANS+LST (Helix) | 0.759 | 0.788 | 0.831 | 0.820 | 0.845 | 0.875 |
| Error [%] | -2.4 | 1.8 | 2.3 | -4.2 | -2.6 | 1.1 |

(b) Medium WS pulse and helix methods

|  | $x/D = 8$ | | | $x/D = 10$ | | |
|---|---|---|---|---|---|---|
|  | Baseline | $A = 2°$ | $A = 4°$ | Baseline | $A = 2°$ | $A = 4°$ |
| AMR-Wind (Pulse) | 0.737 | 0.782 | 0.826 | 0.817 | 0.822 | 0.859 |
| RANS+LST (Pulse) | 0.759 | 0.780 | 0.805 | 0.820 | 0.839 | 0.860 |
| Error [%] | 3.0 | -0.38 | -2.6 | 0.39 | 2.1 | 0.04 |
| AMR-Wind (Helix) | 0.737 | 0.764 | 0.780 | 0.817 | 0.824 | 0.853 |
| RANS+LST (Helix) | 0.759 | 0.788 | 0.831 | 0.820 | 0.845 | 0.875 |
| Error [%] | 3.0 | 3.2 | 6.5 | 0.39 | 2.5 | 2.6 |

(c) High WS helix method

|  | $x/D = 8$ | | | $x/D = 10$ | | |
|---|---|---|---|---|---|---|
|  | Baseline | $A = 2°$ | $A = 4°$ | Baseline | $A = 2°$ | $A = 4°$ |
| AMR-Wind (Helix) | 0.708 | 0.727 | 0.770 | 0.752 | 0.781 | 0.822 |
| RANS+LST (Helix) | 0.701 | 0.703 | 0.706 | 0.706 | 0.711 | 0.722 |
| Error [%] | -0.93 | -3.2 | -8.3 | -6.2 | -8.8 | -12.2 |

for solving these components was formulated. An axisymmetric, parabolized $k$-$\varepsilon$ RANS model was used to solve for the mean flow and fine scale turbulence components. To model the wave components of the flow, a simplified, inviscid, parallel-flow, linear spatial stability analysis was used. The linear stability modes were coupled with the RANS model to capture the interactions between the coherent structures and the mean flow.

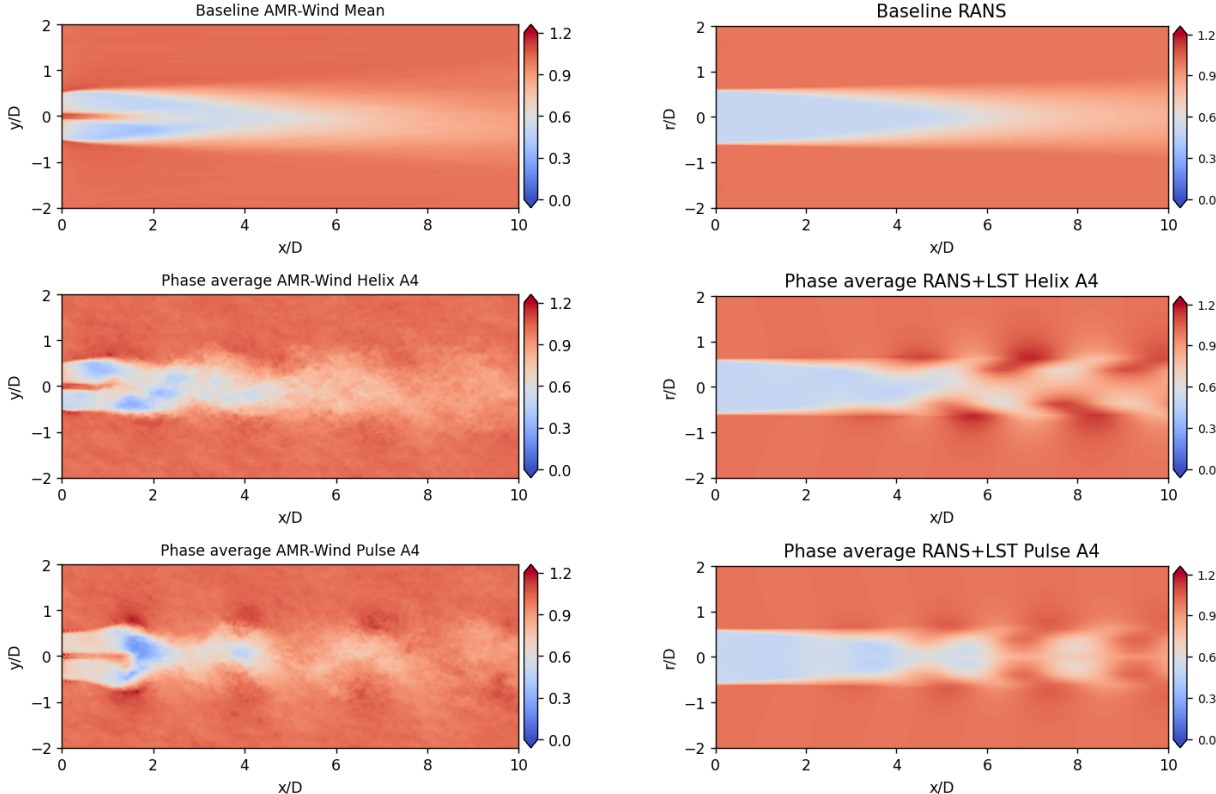

**Figure 13.** Comparisons of the AMR-Wind and RANS with linear stability mean velocity component (top row) with the phase averaged velocity for the HelixA4 (middle row) and PulseA4 (bottom row) cases. In all cases the normalized streamwise velocity, $U/U_\infty$, is plotted for the Low WS case.

Comparisons between the RANS with linear stability model and high fidelity LES calculations of the turbine wakes showed that this framework was able to capture the wake modifications due to AWM actuation, particularly in the far wake regions. Additional wake mixing and more rapid wake recovery was observed for both the pulse and helix AWM strategies. Some differences are also observed in the near wake region of the flow. The high fidelity LES calculations include non-axisymmetric features immediately downstream of the rotor, which the baseline RANS model fails to capture, and the impact of the large-scale coherent structures is also more evident in the near wake region of the LES cases.

There are several limitations associated with the current formulation of the RANS model that could be improved in future studies. One significant constraint is the axisymmetric assumption, which restricts the model's applicability in realistic environments, such as ABLs characterized by large veer. To address this limitation, a parabolic three-dimensional version of the model could be developed using the same principles established here by marching two-dimensional wake profiles downstream, similar to the improvements suggested by Cheung et al. (2024b). Furthermore, incorporating direct interactions between the coherent flow structures and the turbulence may result in a more accurate representation of the flow dynamics, although the

effect on the mean flow from these higher order interactions remains unknown. Lastly, this work has demonstrated the importance of calibrating the RANS model constants; however, further work is needed to establish robust values for these constants, including the use of additional data in the calibration process.

Additional improvements to the linear stability model are also possible. The current model ignores the effects of swirl, shear, and veer, which impacts the growth of the coherent structures and can help improve the comparisons with the high fidelity simulations. It may also be possible to use the full RANS velocity profile in the Rayleigh equation, instead of a piecewise constant approximation, which would help improve near wake predictions. Finally, future work may also investigate the impact of mode-to-mode interactions in a nonlinear stability framework. For example, the interactions between the axisymmetric modes and helical modes may be crucial in determining the optimal forcing strategy, and could be worth exploring in later studies. Finally, additional insight may be gained by comparing the coherent structures' behavior from linear stability theory with modes computed by spectral POD analysis (Yalla et al., 2025). This may provide some indications of which effects are important for the the stability model to capture.

*Code and data availability.* The simulation code details used in this study are available as an attachment to this paper online. This includes the setup input files, code versions, and turbine model details.

*Author contributions.* L. Cheung was responsible for developing the mathematical formulation, model implementation, and manuscript preparation. G. R. Yalla was responsible for the formulation of the reduced order RANS model, model implementation, generation of LES data, and manuscript preparations. M. T. Henry de Frahan was responsible for performance optimization of the RANS model solver, discussions surrounding the AMR-Wind solver, and manuscript contributions. K. Brown was responsible for conceptualization, performing portions of the LES, and manuscript review. P. Mohan was responsible for calibration of the RANS model coefficients, performance optimization of the RANS model solver, and manuscript contributions. A. Hsieh assisted with data post-processing and comparing results. N. deVelder was responsible for the formulation and development of the RANS model. D. Houck assisted with the problem formulation, manuscript review and preparations. M. Day assisted with editing and review of the manuscript and was also responsible for project organization. M. Sprague assisted with editing and was responsible for funding and computer time on OLCF resources.

*Competing interests.* The authors declare that they have no conflict of interest.

*Acknowledgements.* Sandia National Laboratories is a multimission laboratory managed and operated by National Technology & Engineering Solutions of Sandia, LLC, a wholly owned subsidiary of Honeywell International Inc., for the U.S. Department of Energy's National Nuclear Security Administration under contract DE-NA0003525.

This work was authored in part by the National Renewable Energy Laboratory, operated by Alliance for Sustainable Energy, LLC, for the U.S. Department of Energy (DOE) under Contract No. DE-AC36-08GO28308. This research used resources of the Oak Ridge Leadership

Computing Facility at the Oak Ridge National Laboratory, which is supported by the Office of Science of the U.S. Department of Energy

under Contract No. DE-AC05-00OR22725, and under the ALCC allocation "Grand-challenge predictive wind farm simulations".

This material is based upon work supported by the U.S. Department of Energy, Office of Science, Advanced Scientific Computing Research and Biological and Environmental Research programs through the FLOWMAS Energy Earthshot Research Center.

This research has been supported in part by the Wind Energy Technologies Office within the Office of Energy Efficiency and Renewable Energy. The views expressed in the article do not necessarily represent the views of the U.S. DOE or the U.S. Government. This written

work is authored by an employee of NTESS. The employee, not NTESS, owns the right, title and interest in and to the written work and is responsible for its contents. Any subjective views or opinions that might be expressed in the written work do not necessarily represent the views of the U.S. Government. The publisher acknowledges that the U.S. Government retains a non-exclusive, paid-up, irrevocable, world-wide license to publish or reproduce the published form of this written work or allow others to do so, for U.S. Government purposes. The DOE will provide public access to results of federally sponsored research in accordance with the DOE Public Access Plan.

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
