# Peer review of "Modeling the effects of active wake mixing on wake behavior through large-scale coherent structures"

_Wind Energy Science, 2024_

## Referee Comment (RC3)

**Review of manuscript "Modeling the effects of active wake mixing on wake behavior through large scale coherent structures" by Cheung et al.**

The paper proposes a parabolized RANS approach for modeling the effects of turbine-generated flow structures in the wake on recovery based on a triple decomposition approach. The methodology is original, innovative, and pertinent to the growing research community in active wake mixing. However, I believe the presentation of the papers and the analysis of the results could be significantly improved based on the comments below.

**Major comments**

1. Large-eddy simulation data is used to show the agreement of the proposed model to a high-fidelity model. However, the LES setup is insufficiently detailed to allow reproducibility of the results, more specifically the following questions are unanswered.

    a. Section 2.1 mentions that representative conditions are based on floating lidar measurements after a selection process and then Table 1 mentions the resulting WS, TI, etc. obtained from the LES. However, it is unclear how the LES has been set up to match the measurements (which is not a trivial process). Furthermore, an incomplete reference is made to Brown et al. 2025, but I could not find this paper anywhere. Please detail.

    b. The authors mention that the work focuses on larger offshore wind turbines under stable atmospheric conditions (line 70), however it is not discussed whether the LES is a low TI neutral case or effectively a stable case. Details of initialization and precursor setup are important but missing from the manuscript.

    c. The authors mention that AMR-Wind can include mesoscale, Geostrophic, Coriolis, actuator line models etc., but the exact setup used is not detailed.

    d. Is there a reason why the domain lengths are different for different wind speeds?

2. The performance of the RANS vs. LES model in both the baseline and the actuated cases is shown through a qualitative visual comparison of velocity profiles in the form of red and blue lines in Figures 6 - 10. Discrepancies are mostly attributed to the effects of the hub / nacelle and veer / shear in the LES.

a. Considering that inclusion of veer and shear are left for future work, would a comparison to an axisymmetric LES of just the turbine rotor not have facilitated a more direct evaluation of the performance of the current model? Please elaborate why the current approach was chosen.

b. The performance evaluation would be more objective and comprehensive if quantitative numerical error metrics (e.g. MAE, enhanced recovery, …) were introduced. This would facilitate the comparison of performance in different wind conditions as well.

c. Discrepancies between the RANS and the LES are rather large for some of the plots presented, yet they are only very briefly discussed in the text. A somewhat more detailed and objective analysis of the performance of the model would be advised.

3. The parabolized RANS model is described in detail, however some aspects would benefit from further clarification.

a. I was expecting a body force in the momentum equation 5a to represent the turbine force on the flow. Only later, it became clear that the RANS domain only accounts for the region downstream of the turbine. This should be made more explicit in the paper. Does this imply that the current model is limited to the simulation of a single turbine wake? If so, please mention this explicitly, and discuss in more detail practical applicability of the current model.

b. The impact of wave components on the mean field is represented by the term F_CS. The wave field is computed from an analytical linear stability analysis of an axisymmetric piecewise-constant wake profile. However, it is not trivial to understand how the turbine pitch actuations (Table 3) are linked to these modes and hence impact the coupled RANS solution. Are these encoded into the azimuthal wavenumber and temporal frequency of Eq. 15 (and also, is this why the wave component consist of a single exponential basis function rather than a series expansion)? Could this be made more explicit?

**Minor comments**

1. The term large-scale coherent structures is used throughout the paper. In an atmospheric boundary layer context, this term is often used for naturally occurring boundary layer streaks, and their impact on wind turbines and wakes has been studied in several papers in literature (see, e.g. Zhang & Stevens https://doi.org/10.1007/s10546-019-00468-x) To avoid confusion, I would propose to add a disclaimer that the structures here refer to turbine-induced structures in the wake only.

2. Table 1: All cases are Low TI, so why include it in the naming convention? This gives the impression that also medium / high TI cases are included in the investigation. They are also inconsistently referred to throughout the manuscript (e.g. Sometimes as Low WS/Low TI, sometimes as Low WS). This should be simplified.

3. Considering AMR-Wind is a relatively new code, it would be useful for the community to share the setup files for guidance and reproducibility.

4. The definition of the wave component at the bottom of page 4 is implicitly defined. I am assuming this is a typo and the tilde on the right hand side should be omitted.

5. Please introduce all symbols explicitly and uniquely, e.g., $\sigma_k$ and $\sigma_\varepsilon$ are not defined in Eq. 5, $x_0$ is not explicitly defined, $\theta$ is used both as the pitch angle (Eq. 1) and the azimuthal coordinate (Eq. 16), $w$ is never explicitly defined as the azimuthal velocity component (though it is used in the linear stability analysis, etc.

**Typos**

The manuscript contains quite some remaining typos and textual inaccuracies. I list the ones I noted down here below, but expect there are more. Please revise thoroughly.

- Section 2.4.1: constatnt → constant
- Line 261: he wave component → the wave component
- Line 263: the Frobenius norm of (the difference between?) two successive solutions
- Line 265: python → Python
- Line 111: two-dimensions → two dimensions

---

## Author Comment (AC1)

March 6, 2025

We wish to thank the reviewer for their helpful and constructive comments. The reviewer's comments and questions are addressed below. Changes to the manuscript are made in bold red font.

**Reviewer 1**

In this work, the authors proposed an engineering wake model to consider the effects of active wake mixing strategies. The proposed model is based on the triple decomposition of instantaneous velocity into time-averaged components, wave components, and turbulent fluctuations. The time-averaged component is computed by solving the parabolic RANS equations with the turbulent fluctuations modelled using the k-epsilon model. The wave component is modelled using a simplified spatial linear stability formulation. The model predictions are compared with the large-eddy simulation results. An overall good agreement was demonstrated. It is a nice work, allowing fast estimations of different AWM strategies on accelerating the wake flow recovery. Specific comments are as follows:

1. Double check eq. (2), and equations on lines 97 and 99.

   In the revised document, we have fixed typographical errors in the equations, including the definition of the pitch actuation, equation (2), and a few other mistakes in the original manuscript.

2. It is suggested to plot in figure 1: instantaneous, time-average, wave component and turbulent fluctuations.

   As suggested by the reviewer, we have added some additional contour plots to figure 1 (also replicated below) so that it now shows an example of the instantaneous $u(x,t)$, time-average $\overline{U}$, phase average $\langle u \rangle$, wave component $\tilde{u}$, fluctuating component $u'$, and turbulent kinetic energy decomposition of the flow field.

3. Line 261: "he wave".

   This typo has been corrected in the text.

4. The proposed model assumes axisymmetry. In LES the ground is included, making the comparison between the two unfair. It is suggested to run ideal LES cases under uniform inflow and with symmetry BCs on the four sides to verify the model first.

   We agree with the reviewer that, as the LES is currently performed, the comparison could be deemed unfair. A comparison to an axisymmetric LES of just the turbine rotor would have yielded a direct comparison. However, the objective of this work is to provide a usable, proof-of-concept framework that illustrates how a RANS model with a linear stability model can capture most of the phenomena of interest in the LES data. This was adequately demonstrated in the manuscript. The long-term goal, as

[Figure]

**Figure 1.** An example of a triply decomposed flow field for a wind turbine wake. This case is from the HelixA4 case under stable Low WS ABL conditions with $4°$ amplitude forcing. In each contour, the normalized streamwise velocity, $U/U_\infty$, is plotted.

stated in the conclusions discussing future work, of such a framework is to build it up from common principles towards being able to capture increasing physics complexity, such as shear effects, veer, and asymmetry. As such, it does not serve the current manuscript to remove physical phenomena and perform an ideal LES with uniform inflow and symmetry boundary conditions. This would be discarding the important physics of wind farm LES without informing the potential failure modes of the proposed framework to point to future improvements. Given the simplicity of the resulting flow of an axisymmetric LES without boundary conditions, it can be fully expected that a calibrated axisymmetric RANS model as proposed in the manuscript would fully be able to capture the physical quantities of interest. The current work has the merit of highlighting that the current approach performs well in comparison with complex LES data while also highlighting future improvements.

Though noted in the conclusions, other parts of the manuscript do not clearly lay out these goals and, therefore, additional discussion along these lines was added to Section 3.1.

5. Discrepancies shown in figure 9 are large. Discussions are necessary.

Additional discussion regarding the observed discrepancies for the High WS case are now included in section 3.1 and 3.2 of the revised manuscript. In section 3.1, we show that mean flow calculated using the RANS model overestimates the potential core region of the wake when compared to the equivalent AMR-Wind case, leading to lower

centerline and rotor averaged velocity. These differences then impact the behavior of the large-scale coherent structures as predicted by the RANS and linear stability model. The larger potential core region in the RANS profiles limits the modifications from the coherent structures to the wake shear regions until farther downstream in the wake. This leads to relatively minor changes to the centerline velocity for the High WS case compared to the Med or Low WS cases. These results suggest that accurately capturing the mean flow is critical to modeling the impact of large scale structures on wake behavior. We hope that future work in this area will improve the baseline RANS models and calibrations, and increase the accuracy of the overall model.

6. There are models in the literature developed to predict coherent flow structures in wind turbine wakes (e.g., J. Fluid Mech. (2024), vol. 980, A48, doi:10.1017/jfm.2023.1097). It is suggested to review them in the introduction section.

A paragraph has been added to the introduction that discusses dynamic wake models, in addition to the steady-state models that are commonly used for wind farm optimization. The primary focus is on data-driven representations of coherent flow structures. Three recent developments using Spectral Proper Orthogonal Decomposition, Resolvant Analysis, and Dynamic Mode Decomposition are discussed in the context of Active Wake Mixing. The Dynamic Wake Meandering model is also briefly mentioned as another reduced-order modeling approach aimed at representing unsteady dynamics.

---

## Author Comment (AC2)

March 6, 2025

The authors wish to thank the referee for the time and effort spent in reviewing the manuscript. The comments have helped to improve the paper and clarify important points that we discuss. Ours responses to the comments are included below.

**Reviewer 2**

The paper aims to develop a framework for modeling active wake farm mixing, with particular attention to the impacts of large scale coherent structures and turbulence on the mean flow. The model is interesting and provide a new way to analyze a promising approach for reducing wake effects. However, the literature review is incomplete and focuses on models not designed to capture the features of interest and other attempts to investigate coherent structures in the wakes, which actually makes the paper claims less compelling. Detailed comments regarding this point as well as other minor comments/questions follow.

1. The introduction and comparisons focus on the improvement of the model with respect to static wake models, which are not designed to capture dynamic behavior. There are a number of dynamic models that would serve as a better focus of both the literature review and comparisons.

    A paragraph has been added to the introduction to discuss dynamic wake models alongside the steady-state models that are commonly used for wind farm optimization. The primary focus is on data-driven representations of coherent flow structures, including models based on Proper Orthogonal Decomposition (POD), Resolvent Analysis, and Dynamic Mode Decomposition (DMD). The Dynamic Wake Meandering model is also briefly mentioned as another reduced-order modeling approach aimed at capturing unsteady dynamics.

2. Resolvant analysis has been recently applied to study wind farm wakes and it would be useful to compare this approach (or at least include it the literature review), i.e. on the top of page 3 where the authors mention that large-scale coherent structures have not been studied in this context. There have also been POD and DMD based studies of wind farm wakes that precisely aim to characterize coherent structures in wind farm wakes. DMD is in fact a dynamic approach.

    As mentioned in our response to the previous comment, we have added a paragraph on dynamic wake modeling to the introduction, focusing on data-driven representations of coherent flow structures. Three recent developments using Spectral SPOD, Resolvent Analysis, and DMD are discussed in the context of Active Wake Mixing. Additionally, we discuss the limitations of data-driven approaches and our rationale for pursuing an analytical representation of coherent flow structures instead.

3. The paper mentions the focus being on offshore and stable atmospheric conditions (line 70) but none of the results and model development are applicable to stable conditions.

This point should be clarified, in fact I suggest removing this statement since it does not accurately reflect the paper content (which clearly states the linear stability analysis does not include key effects of a stable boundary layer line 170)

We agree with the reviewer that the mention of stability conditions is not reflective of the current capabilities of the RANS and linear stability model. In the revised manuscript, we have clarified this statement so that it now refers to low TI conditions, which would be the likely environment where AWM could be applied.

4. Why is RANS the best approach for this work? Many RANS closure models are known to have some limitations simultaneously capturing both the mean flow and wave behavior and it would be useful to understand how/why the configuration selected overcomes these issues and why this approach is better than the alternatives.

We adapt a RANS approach since the primary goal for our model is to capture the effects of the coherent structures on the mean flow with a spatial linear stability formulation. Our results show that we can accomplish with the triple decomposition formulation as described in sections 2.2 and 2.3 where the large-scale coherent structures can impact the mean flow and vice-versa. We also want to note that our approach is neither recommending nor is limited to the $k - \epsilon$ turbulence model; rather, it represents one common approach to modeling the effects of turbulence. Our formulation is generalizable to most common RANS closure models.

5. There are a number of grammatical issues (another careful proof reading is likely to catch these)

The revised manuscript has been reviewed for grammatical errors and typos and several corrections have been made.

6. The coordinate frame should be specified. The authors use y and r, clearly different coordinate frames are used, so clarification would be useful.

In the revised manuscript, we have included a schematic in figure 2 (shown below) that shows both the cylindrical coordinate system used in this work, and its relationship to the Cartesian system relative to a typical turbine.

[Figure]

7. In many shear flows, singular values (e.g. resolvant modes or POD modes) provide more accurate characterization of the behavior of coherent structures, why are eigenvalues the best approach here?

Both the singular value decomposition and the eigenvalue decomposition offer useful representations of data, and they are often related. For instance, the reviewer mentions POD modes in connection to singular values but, in fact, the standard (space-only) POD is given by the solution to the Fredholm eigenvalue problem:

$$\int_{\Omega} \boldsymbol{C}(\boldsymbol{x}, \boldsymbol{x}')\boldsymbol{\phi}(\boldsymbol{x}')d\boldsymbol{x}' = \lambda\boldsymbol{\phi}(\boldsymbol{x}), \tag{1}$$

where $\boldsymbol{C}(\boldsymbol{x}, \boldsymbol{x}')$ is the two-point spatial correlation tensor. The solution to (1) is a set of eigenvectors, $\boldsymbol{\psi}$, and eigenvalues, $\lambda$, that represent the coherent flow structures and the average TKE captured by the flow structures, respectively. Therefore, eigenvalues are naturally associated with the coherent structures in a flow. A connection to singular values arises when solving (1) discretely. The analytical eigenvalue problem is represented discretely by a system of the form $\mathbf{C}\boldsymbol{\psi} = \lambda\boldsymbol{\psi}$, where $\mathbf{C} = \mathbf{U}\mathbf{U}^H$ and $\mathbf{U} \in \mathbb{R}^{N_x \times N_t}$. Often, the number of points in time ($N_t$) that are used to form $\mathbf{C}$ is much less than the number of spatial points ($N_x$), and so this system is solved efficiently by performing a low-rank SVD of the matrix $\mathbf{U} = \mathbf{L}\boldsymbol{\Sigma}\mathbf{R}^H$. Using this decomposition, $\mathbf{C}$ can be expressed as $\mathbf{C} = \mathbf{L}\boldsymbol{\Sigma}^2\mathbf{L}$. Thus, the eigenvalues, $\lambda$, are given by the square of the singular values and the eigenvectors, $\boldsymbol{\psi}$, are given by the left singular vectors.

The connection between eigenvalues and singular values extends beyond computational efficiency. As the reviewer points out, the optimal inputs and outputs to the resolvent operator, $\mathcal{R}$, in a Resolvent-analysis is defined through the SVD, $\mathcal{R} = \sum_j \sigma_j \mathbf{u}_j \otimes \mathbf{v}_j$. Here, $\mathbf{v}_j$ are the input modes and $\mathbf{u}_j$ are the output modes, and the "gain" between input and output pairs are quantified by the square of the singular value $\sigma_j^2$. A connection between resolvent analysis and (Spectral) POD is obtained by expressing

the Fourier transform of the two-point correlation tensor in terms of resolvent modes. In this case, the POD eigenvalues are found to be the square of the resolvant singular values (see Towne, Schmidt, and Colonius (2018) for more details).

In the work here, we focus on eigenvalues because they directly inform us about the stability characteristics of flow structures. Specifically, the solution to the Rayleigh equation formulated in Section 2.4.2 is obtained by solving the corresponding characteristic equation assuming a solution of the form $\tilde{\phi}(x, r, \psi, t) = \hat{\phi}_n(r)e^{i\alpha x + in\psi - i\omega t}$. Here, $\alpha$ are the eigenvalues that are associated with the linear operator defined by the differential equation. The real part of the eigenvalue determines the wavelength of the large-scale coherent structures, while the imaginary component of the eignenvalue quantifies the spatial growth of the structures.

---

## Author Comment (AC3)

March 6, 2025

The authors are very appreciative of the reviewer's time and efforts in evaluating the manuscript. We have made several improvements and clarifications to the manuscript due to the suggestions that were received. Please find our responses to the earlier comments below.

**Reviewer 3**

The paper proposes a parabolized RANS approach for modeling the effects of turbine generated flow structures in the wake on recovery based on a triple decomposition approach. The methodology is original, innovative, and pertinent to the growing research community in active wake mixing. However, I believe the presentation of the papers and the analysis of the results could be significantly improved based on the comments below.

**Major Comments**

1. Large-eddy simulation data is used to show the agreement of the proposed model to a high-fidelity model. However, the LES setup is insufficiently detailed to allow reproducibility of the results, more specifically the following questions are unanswered.

    (a) Section 2.1 mentions that representative conditions are based on floating lidar measurements after a selection process and then Table 1 mentions the resulting WS, TI, etc. obtained from the LES. However, it is unclear how the LES has been set up to match the measurements (which is not a trivial process). Furthermore, an incomplete reference is made to Brown et al. 2025, but I could not find this paper anywhere. Please detail.

    The authors apologize for the incomplete reference to Brown et al. (2025). This reference has been properly completed and the manuscript in review can be found here: `https://wes.copernicus.org/preprints/wes-2024-191/`

    (b) The authors mention that the work focuses on larger offshore wind turbines under stable atmospheric conditions (line 70), however it is not discussed whether the LES is a low TI neutral case or effectively a stable case. Details of initialization and precursor setup are important but missing from the manuscript.

    Yes, thank you for pointing out these omissions. In addition to the reference to Brown et al. (2025) for more details, we have added the following sentences: *To generate the precursors, small velocity and temperature perturbations were introduced near the surface to accelerate turbulence development. The low-TI conditions were produced by imposing negative ground surface temperature rates and adjusting the surface roughnesses, followed by 10000s of flow time. As such, the generated conditions were stable atmospheric boundary layers.*

    (c) The authors mention that AMR-Wind can include mesoscale, Geostrophic, Coriolis, actuator line models etc., but the exact setup used is not detailed.

We have updated the language to specify which of the types of available forcing are actually used in this study: *AMR-Wind includes all the necessary physics modules to simulate atmospheric boundary layers (ABLs); included in this effort are ABL forcing, Boussinesq buoyancy, Coriolis forcing, body forcing to maintain the precursor-derived inflow condition in the presence of the turbine's blockage, and body forcing from coupling to OpenFAST for turbine representation using actuator line models (these are the same forcing terms used in Brown et al. [2025] and Hsieh et al. [2025], for instance).*

In addition, we have provided the input files for all the simulations as Supplemental Material so that they are archived with this manuscript.

(d) Is there a reason why the domain lengths are different for different wind speeds?

There is no significant reason behind the different domain sizes between the Med WS and Low WS/High WS cases. This difference was an artifact of evolving test goals, however, the smaller domain is not believed to meaningfully impact the results since the outflow plane is still $> 13$ rotor diameters from the turbine for the shorter domain.

2. The performance of the RANS vs. LES model in both the baseline and the actuated cases is shown through a qualitative visual comparison of velocity profiles in the form of red and blue lines in Figures 6 - 10. Discrepancies are mostly attributed to the effects of the hub / nacelle and veer / shear in the LES.

(a) Considering that inclusion of veer and shear are left for future work, would a comparison to an axisymmetric LES of just the turbine rotor not have facilitated a more direct evaluation of the performance of the current model? Please elaborate why the current approach was chosen.

We agree with the reviewer that, as the LES is currently performed, the comparison could be deemed unfair. A comparison to an axisymmetric LES of just the turbine rotor would have yielded a direct comparison. However, the objective of this work is to provide a usable, proof-of-concept framework that illustrates how a RANS model with a linear stability model can capture most of the phenomena of interest in the LES data. This was adequately demonstrated in the manuscript. The long-term goal, as stated in the conclusions discussing future work, of such a framework is to build it up from common principles towards being able to capture increasing physics complexity, such as shear effects, veer, and asymmetry. As such, it does not serve the current manuscript to remove physical phenomena and perform an ideal LES with uniform inflow and symmetry boundary conditions. This would be discarding the important physics of wind farm LES without informing the potential failure modes of the proposed framework to point to future improvements. Given the simplicity of the resulting flow of an axisymmetric LES without boundary conditions, it can be fully expected that a calibrated axisymmetric RANS model as proposed in the manuscript would fully be able to capture the physical quantities of interest. The current work has the merit of highlighting

that the current approach performs well in comparison with complex LES data while also highlighting future improvements.

Though noted in the conclusions, other parts of the manuscript do not clearly lay out these goals and, therefore, additional discussion along these lines was added to Section 3.1.

(b) The performance evaluation would be more objective and comprehensive if quantitative numerical error metrics (e.g. MAE, enhanced recovery, ...) were introduced. This would facilitate the comparison of performance in different wind conditions as well.

In table 4 and section 3.2 of the revised manuscript, we have now included quantitative error comparisons between the RANS with linear stability model and the AMR-Wind LES calculations. Table 4 compares the hub-height streamwise velocity, i.e., the maximum wake deficit, at the downstream positions of $x/D=8$ and 10. The results are consistent with earlier wake profile comparisons in the manuscript: For the Med and Low WS cases with helix and pulse forcing, the majority of the velocity errors are below 5%, and as expected, the largest differences compared to AMR-Wind occurred for the High WS cases.

(c) Discrepancies between the RANS and the LES are rather large for some of the plots presented, yet they are only very briefly discussed in the text. A somewhat more detailed and objective analysis of the performance of the model would be advised.

Additional material has now been included regarding the differences between the RANS and the LES results. In section 3.1, we discuss the comparisons for the high WS case, and note that the potential core region is overestimated in the RANS model, while it is correctly modeled for the Low and Medium WS case. This leads to discrepancies in both the centerline velocities and rotor averaged averaged velocities. This discrepancy in the High WS RANS model also impacts the later comparisons when the large scale structures are also included (section 3.2). We believe that these discrepancies can be reduced through improvements in the RANS model and additional calibration across a wider variety of wake cases in future work.

3. The parabolized RANS model is described in detail, however some aspects would benefit from further clarification.

(a) I was expecting a body force in the momentum equation 5a to represent the turbine force on the flow. Only later, it became clear that the RANS domain only accounts for the region downstream of the turbine. This should be made more explicit in the paper. Does this imply that the current model is limited to the simulation of a single turbine wake? If so, please mention this explicitly, and discuss in more detail practical applicability of the current model.

The reviewer has raised an excellent point regarding the applicability of the current model. This point has now been clarified in section 2.2 of the manuscript, where we now state that the RANS and linear stability model applies to the wake of a single turbine immediately downstream of the rotor. The inflow and the rotor dynamics are not included in this formulation, and the behavior of more complicated phenomena, such as the merging of multiple wakes, is not considered in this work. However, the intention of the authors is to extend this model in future work so that it can be used for wind farm configurations with multiple turbines. As discussed in the response above, effects like shear, veer, and other flow asymmetries need to be developed first, after which it can then be applied to more complicated configurations.

(b) The impact of wave components on the mean field is represented by the term $F_{CS}$. The wave field is computed from an analytical linear stability analysis of an axisymmetric piecewise-constant wake profile. However, it is not trivial to understand how the turbine pitch actuations (Table 3) are linked to these modes and hence impact the coupled RANS solution. Are these encoded into the azimuthal wavenumber and temporal frequency of Eq. 15 (and also, is this why the wave component consist of a single exponential basis function rather than a series expansion)? Could this be made more explicit?

In section 2.6 of the revised paper, we present more details regarding the blade pitch actuation parameters and the instability modes used in the current analysis. The connection between the two was discussed more fully in Cheung et al (*Energies*, 2024), but the relevant details are included here for completeness. In that study, different blade pitch actuation strategies were applied to an OpenFAST turbine model simulation using different pitch amplitudes, azimuthal mode numbers, and the desired Strouhal frequency St=0.30. An analysis of the resulting blade loads showed that there was a corresponding fluctuating streamwise blade force that appeared at the same azimuthal mode number and Strouhal frequency. Furthermore, through a spectral POD analysis, we can see that these fluctuating streamwise blade forces then excite a similar response in the near wake (see Yalla et al, 2025). While the radial distribution of the fluctuating blade forces due to the AWM actuation strategy may not exactly match the eigenfunction solutions of equations 22, we believe that it is sufficient to pitch the blades at the specified azimuthal mode number $n$ and temporal frequency $\omega$ (or Strouhal number St) to excite the desired instability mode.

The reviewer is also correct in noting that this study considered the impact of a single instability wave, at a single Strouhal number and a specific azimuthal mode number. In the more general case, multiple instability wave components can be included in the analysis, and a summation over all wave components in equation 15 is then required. This would allow for AWM strategies such as the side-to-side actuation to be analyzed, or for the behavior of the higher harmonics to be included in the wake model. However, because nonlinear interactions among

instability modes is not within the scope of the current analysis, only a single mode is considered here. This additional clarification is now also explicitly included in section 2.4.2.

**Minor Comments**

1. The term large-scale coherent structures is used throughout the paper. In an atmospheric boundary layer context, this term is often used for naturally occurring boundary layer streaks, and their impact on wind turbines and wakes has been studied in several papers in literature (see, e.g. Zhang & Stevens https://doi.org/10.1007/s10546-019-00468-x) To avoid confusion, I would propose to add a disclaimer that the structures here refer to turbine-induced structures in the wake only.

   The introduction has been updated to clarify that the focus is on modeling coherent structures generated from the turbine. Moreover, it is mentioned that large-scale coherent structures can arise from other sources, such as the naturally occurring boundary layer streaks in an atmospheric boundary layer, which also affect wake dynamics, and that the modeling framework developed in the paper may be relevant for these processes as well.

2. Table 1: All cases are Low TI, so why include it in the naming convention? This gives the impression that also medium / high TI cases are included in the investigation. They are also inconsistently referred to throughout the manuscript (e.g. Sometimes as Low WS/Low TI, sometimes as Low WS). This should be simplified.

   We simplified the naming of the cases to remove all mention of Low TI since all of the cases were run with the same TI.

3. Considering AMR-Wind is a relatively new code, it would be useful for the community to share the setup files for guidance and reproducibility.

   The public AMR-Wind documentation contains extensive code and user documentation. Of particular interest to the users are a set of walk-through documents that provide concrete, detailed input files for a range of cases. In addition, we have provided the input files for all the simulations as Supplemental Material so that they are archived with this manuscript.

4. The definition of the wave component at the bottom of page 4 is implicitly defined. I am assuming this is a typo and the tilde on the right hand side should be omitted.

   This typo, and other similar typographical mistakes, have been corrected in the revised manuscript.

5. Please introduce all symbols explicitly and uniquely, e.g., $\sigma_k$ and $\sigma_\epsilon$ are not defined in Eq. 5, $x_0$ is not explicitly defined, $\theta$ is used both as the pitch angle (Eq. 1) and the azimuthal coordinate (Eq. 16), $w$ is never explicitly defined as the azimuthal velocity component (though it is used in the linear stability analysis, etc.

In the revised manuscript, we have corrected several mistakes and clarified the mathematical notation. This includes:

- Describing $C_{1\varepsilon}$, $C_{2\varepsilon}$, $C_\mu$, $\sigma_k$ and $\sigma_\varepsilon$ in the RANS model equations as calibration constants and defined in Sec. 2.3.1.

- Providing an explicit definition of $x_0$.

- Including a schematic of the coordinate system used in figure 2, and consistently using $\psi$ as the azimuthal coordinate.

- Defining $w$ as the azimuthal velocity

- The azimuthal mode number is now consistently defined as $n$.

**Typos**

The manuscript contains quite some remaining typos and textual inaccuracies. I list the ones I noted down here below, but expect there are more. Please revise thoroughly.

1. Section 2.4.1: constatnt - constant

2. Line 261: he wave component - the wave component

3. Line 263: the Frobenius norm of (the difference between?) two successive solutions

4. Line 265: python - Python

5. Line 111: two-dimensions - two dimensions

These typos and grammatical errors have been corrected in the revised manuscript